# Air clathrate hydrates in the EDML ice core, Antarctica

Florian Painer<sup>1,2</sup>, Sepp Kipfstuhl<sup>1</sup>, Martyn Drury<sup>3</sup>, Tsutomu Uchida<sup>4</sup>, Johannes Freitag<sup>1</sup>, and Ilka Weikusat<sup>1,2</sup>

<sup>1</sup>Alfred-Wegener-Institut Helmholtz-Zentrum für Polar- und Meeresforschung, Bremerhaven, Germany

<sup>2</sup>Department of Geosciences, Eberhard Karls University, Tübingen, Germany

<sup>3</sup>Department of Earth Sciences, Utrecht University, The Netherlands

<sup>4</sup>Division of Applied Physics, Faculty of Engineering, Hokkaido University, Sapporo, Japan

**Correspondence:** Florian Painer (florian.painer@awi.de)

**Abstract.** In the deeper part of polar ice sheets, air hydrates trap most of the ancient air molecules, which are essential for understanding past climate. We use digital image analysis to create a high-resolution record of air hydrate number, size and shape from ice thick section microphotographs of the EPICA Dronning Maud Land (EDML) ice core, Antarctica, over a depth range from 1255 - 2771 m. We confirm that the air hydrate number and size correlate with paleoclimate and that the correlation disappears in the deeper parts of the ice core, which was previously shown for the Vostok and Dome Fuji ice cores in Antarctica and the GRIP ice core in Greenland. We also observe that the air hydrates grow with depth. Furthermore, we identify two peculiarities: A distinctive change in air hydrate aspect-ratio and orientation at about 2030 m and a region of increased air hydrate clustering from 2392 - 2545 m depth. Remarkably, they coincide with regions of distinctive changes in ice microstructure as response to changes in local ice dynamics and, therefore, we discuss the influence of ice deformation on the air hydrate ensemble.

Copyright statement. TEXT

#### 1 Introduction

Polar ice cores are a unique climate archive, as they provide the most direct record of past atmospheric greenhouse gas compositions (e.g., Raynaud et al., 1993). In the shallower parts of polar ice sheets, below the firn to ice transition, air bubbles are the main type of air inclusions. In the bubble to hydrate transition zone (BHTZ), air bubbles gradually transform to clathrate hydrates of air (hereinafter, air hydrates). Air hydrates are a guest-host compound with a crystalline framework of hydrogen-bonded water molecules forming cages (host) in which individual air molecules (guests) are trapped (e.g., Chazallon and Kuhs, 2002). X-ray diffraction studies on single crystals of air hydrates in the Dye-3 and Vostok ice cores revealed the Stackelberg's crystallographic structure II (sII; Hondoh et al., 1990; Takeya et al., 2000). At standard temperature and pressure, one volume of air hydrate can store up to 150 volumes of air (Uchida et al., 2011). Therefore, after the completion of the bubble to hydrate transformation, air hydrates are considered to trap most of the ancient air molecules in polar ice (Uchida et al., 2011). Exten-

sive records of air hydrate number concentration ( $N_{ah}$ ), size and shape were acquired for the Vostok (Uchida et al., 1994a; Lipenkov, 2000) and Dome Fuji (Narita et al., 1999; Ohno et al., 2004; Uchida et al., 2011) ice cores in Antarctica and the GRIP ice core in Greenland (Pauer et al., 1999) using optical light microscopy. These investigations discovered, that the air hydrates'  $N_{ah}$  and mean size correlate with past climatic changes. Generally, ice formed during colder conditions is characterized by a higher  $N_{ah}$  of smaller air hydrates, while ice formed during warmer conditions has a lower  $N_{ah}$  of larger air hydrates (e.g., Uchida et al., 1994a; Pauer et al., 1999). It is suggested that this relation originates in the formation of air bubbles at the firn/ice transition, which is connected to the grain size of the ice matrix and, therefore, to the temperature and accumulation rate prevailing during the snow to ice transformation (Lipenkov, 2000; Spencer et al., 2006; Lipenkov and Salamatin, 2014; Lipenkov, 2018). Higher temperatures and lower accumulation rates lead to bigger grains and, as a result, to bigger, but fewer air bubbles. Despite the intricate processes that govern the conversion of air bubbles to air hydrates, the correlation between the number concentration and size of air inclusions with different climatic conditions persists after the BHTZ (Uchida et al., 1994a; Lipenkov, 2000). With increasing depth, the air hydrate ensemble evolves due to physicochemical processes such as a) fragmentation and recrystallization (Kipfstuhl et al., 2001) and b) air molecule diffusion and subsequent crystal growth (e.g., Salamatin et al., 2003; Uchida et al., 2011) occurring within the ice sheet.

The EPICA Dronning Maud Land (EDML) deep ice core was drilled at Kohnen Station in East Antarctica (79°00′ S, 0°04′ E, elevation 2892 m .a.s.l.) from the year 2001 to 2006. The maximum logging depth is 2774.15 m (Wilhelms et al., 2014) and the borehole inclination is roughly between ± 3° (Weikusat et al., 2017). It differs from other Antarctic deep ice cores from the East Antarctic Plateau by a current higher accumulation rate (about 7 cm a<sup>-1</sup> ice equivalent) and higher annual average surface temperatures (-45 °C), which allowed obtaining paleoclimatic records at a high temporal resolution (e.g., EPICA Community Members, 2006). It is dated down to about 2415 m depth, or 145 kyr (Ruth et al., 2007; Bouchet et al., 2023) and the BHTZ extends from 700 m to 1225 m (Bendel et al., 2013). The ice cores' location on an ice divide and the possibility to analyze air hydrates' properties with a higher temporal resolution, compared to the Dome Fuji and Vostok ice cores, provide an interesting and novel case for the study of air hydrate inclusions.

Typically, the quantification of air hydrates and their properties in polar ice required tedious and time-consuming manual measurements - approximately 200 inclusions per sample were characterized to obtain statistically representative values. In the past years, digital optical microscopy and automatic image analysis were employed to characterize ice grains and grain boundary structures (Kipfstuhl et al., 2009; Binder et al., 2013) as well as air bubble inclusions (Ueltzhöffer et al., 2010; Bendel et al., 2013). Automated image analysis offers advantages compared to manual methods, such as repeatability and analysis speed, and allows the investigation of larger datasets.

In this work, for the first time, we apply traditional image analysis algorithms on ice thick section microphotographs to analyze air hydrates in an ice core. Altogether, we measure and evaluate about 183,000 air hydrates in 110,000 images, which belong to 74 ice thick section samples from the EDML ice core (Kipfstuhl, 2007). The aim of this contribution is to provide a high-resolution record of air hydrate properties below the BHTZ of the EDML ice core. We investigate the correlation of the air hydrate properties and past climate at higher resolution than previously possible for other Antarctic ice cores. In addition, we examine the air hydrate evolution via depth.

#### 2 Data and Methods

#### 2.1 Data-set: EDML thick-section images

In the course of the EDML microstructure mapping project (Kipfstuhl et al., 2006; Kipfstuhl, 2007), microphotographs of vertical ice thick sections ( $\sim 90 \text{ x } 45 \text{ x } 4.5 \text{ mm}$ ) of the EDML ice core were acquired every 10 m depth. For this study, we selected microphotographs of 74 thick sections in the depth range of 1255 m - 2774 m, i.e. below the BHTZ at EDML. The images are unique, because they were recorded during three Antarctic field seasons (2002 - 2006), within a few days after the sample was drilled. An exception are ten microphotographs from 1314 m to 1554 m, which were obtained 5 months after drilling. Air hydrates decompose during the long time storage of the ice samples (Uchida et al., 1994b), therefore, the imaging as soon as possible after core retrieval is crucial. The microstructure mapping-setup consisted of an optical microscope (Leica DMLM), a CCD video camera (Hamamatsu C5405), a frame grabber and a software-controlled xy-stage (Kipfstuhl et al., 2006). Images were taken in transmission focusing on the surface and into the sample about one- and two-thirds below the surface. Recording of one sample results in a mosaic image, which is composed out of approximately 1500 individual images (tiles), each consisting of 768 x 512 pixels with resolution of 3.25  $\mu$ m per pixel. For this study, mosaic images located about 1.5 mm below the sample surface were analyzed. Sample preparation and image acquisition (microstructure mapping) are described in detail in Kipfstuhl et al. (2006).

# 2.2 Optical and imaging properties of air hydrates

Uchida et al. (1995) measured the mean refractive index of an air hydrate crystal in polar ice and determined it to be 1.3137 ( $\pm$  0.0016), which is slightly higher than the refractive index of ice (1.3084  $\pm$  0.0007). The higher refractive index of air hydrates was also confirmed by the Becke-Test (Shoji and Langway, 1982). Reflection and refraction at the ice / air hydrate boundary focus the light locally to form a bright rim (Becke-Line) and a corresponding shadow creating different "core" and "rim" appearances (A-F in Fig. 1). As a consequence, air hydrates appear in the images with varying optical characteristics, depending on the location to the focus plane. Hydrates close to the plane of perfect focus (or the depth of field) (A, C in Fig. 1) show a small "rim" and a "core" with a brightness close to the pure ice background. Hydrates above the focus plane are characterized by a distinctly darker "core", and a bright "rim" (E in Fig. 1) and hydrates below the focus plane by a brighter "core" and a darker "rim" (B, D in Fig. 1). Hydrates which are outside of the "hydrate-mapping depth" (F in Fig. 1) are hardly, or not at all, distinguishable from the ice background or image artifacts (III in Fig. 1). The presence of these hydrates can only be confirmed by comparing images of different focus levels of the same sample (c.f. section 2.1). Qualitatively, the mean gray value of the hydrate's core as well as the thickness of the rim depend on the distance to the focus plane. Furthermore, other optical features such as secondary formed air bubbles (I in Fig. 1), grain boundaries (II, IV in Fig. 1) and unidentifiable inclusions (V in Fig. 1) are present in the images as artifacts.

**Figure 1.** This figure ("horizontal microscope") shows the different visual categories (A-F) of air hydrates present in the image-dataset, which depend on their relation to the focus plane, as well as other optical features (I-V). The Z-axis is parallel to the core axis and points towards the "top" direction.

### 2.3 Image processing

Python 3 (version: 3.10.13) together with the open-source libraries NumPy (2.1.3), scikit-image (0.24.0), scikit-learn (1.5.2) and Pandas (2.2.3) are utilized for the image and data processing workflows. The routines were implemented on a HP EliteBook 850 G8 with an Intel Core-i5 CPU and 32GB physical memory. The image data can be processed and analyzed within a day. Our image processing and analysis workflow consists of four steps: 1) image stitching, 2) segmentation and filtering, 3) connected component analysis and 4) a plausibility check (Fig. 2).

## 2.3.1 Image stitching

We used a semi-automatic stitching script and fixed offset parameters to merge the 1500 individual tiles into a single mosaic image in a "snake-by-rows" way (Fig. 2). The optimal offset parameters for each of the 74 mosaic images were determined manually. To remove artifacts at the samples edges, the mosaic images were cropped into 80 x 30 mm. Then, they were manually sorted into six different categories depending on acquisition time (i.e. field season), brightness and contrast conditions for further image segmentation (Fig. A1).

#### 2.3.2 Air hydrate segmentation and filtering

Segmentation separates regions with specific characteristics within an image from a background (e.g. air hydrates from an ice background). Objects and background in an image are divided by edges, which are rapid local changes in image color, intensity or texture (Jing et al., 2022; Szeliski, 2022). We use the Canny edge detection algorithm (Canny, 1986), which is one of the most popular edge detection methods (Jing et al., 2022), to produce a binary edge map (Fig. 2b). Then, we apply a combination of binary morphological operators (i.e. dilation and erosion) and selective filtering to obtain the final segmentation result where

Figure 2. Image processing and analysis workflow.

air hydrates correspond to white- and the background to black pixels (Fig. 2c). The individual steps of the segmentation routine are described in detail in Appendix B1. Due to different imaging conditions (i.e. contrast and brightness), which were caused by, among other things, the image acquisition during different Antarctic field seasons, the performance of the edge detection algorithm may vary. Accordingly, we slightly adjusted the segmentation parameters to each of the six image categories by optimizing the segmentation metrics for several manually segmented images, hereinafter ground truth images (see section 2.4 for details).

#### 2.3.3 Analysis of air hydrate properties

In the final segmented image, each object is assigned a unique label and henceforth considered as one air hydrate (Figure 2d). Analyzed pristine properties include the area (i.e. amount of pixels =  $A_{ah}$ ), perimeter and mean gray value from which further parameters were calculated (see equations 1-3). We defined the size of each air hydrate as the equivalent diameter (D) of a circle having the same  $A_{ah}$  as the object's projection.

$$D = 2 \cdot \sqrt{\frac{A_{ah}}{\pi}} \tag{1}$$

The volume (V) of an air hydrate is estimated using a spherical approximation.

$$V = \frac{\pi(D)^3}{6} \tag{2}$$

The circularity is a measure of the similarity of an object to a circle, where 1 denotes a perfect circle and 0 a highly non-circular shape.

$$circularity = \frac{4\pi \cdot A_{ah}}{perimeter^2} \tag{3}$$

In order to analyze the aspect ratio (AR) and orientation, we fitted an ellipse to each air hydrate using scikit-image (van der Walt et al., 2014). We defined the air hydrates' AR as the ratio of the major- to the minor axis of the fitted ellipse and the orientation as the angle ( $\alpha$ ) between the semi-major axis and the x-axis of the microphotographs. The measured values for  $\alpha$  range from 90° to -90° ( $\equiv$  N to S on a compass), where 0° represents a preferred elongation parallel to the image x-axis or, in other words, normal to the core axis (geographic horizontal). For the evaluation of the orientation data, we considered only those air hydrates whose ratio of major- to minor axis is higher than 4:3 (i.e. an AR higher than 1.33).

# 2.3.4 Cleaning sample-specific noise (plausibility check)

We examined two-dimensional histograms of the diameter against circularity (Fig. 2), AR and mean gray value for each air hydrate in a sample in order to perform a plausibility check of the segmentation. This method allows for a fast visual assessment by inspecting and comparing the resulting patterns of the 2D histograms. For samples from 1314 m, 2055 m and 2115 m depth, an additional filter, based on the results of the plausibility check, was applied to remove sample-specific noise. For instance, for the sample from 1314 m, small and round (small diameter and high circularity) objects were removed from the dataset (Fig. 2).

### 2.4 Segmentation evaluation and metrics

125

135

To evaluate the segmentation routine, we created eight ground truth (GT) images with the dimensions of 20 x 10 mm. They were selected to represent the six image categories and to reflect the diversity of the dataset (Fig. A1). Each GT image contains 160 - 387 air hydrates which were traced by hand and classified into five visual categories using the ImageJ / Fiji plugin LABKIT (Arzt et al., 2022). Emphasis was placed on labeling an image in one run and the individual images in quick succession in order to keep human bias as low as possible. The labeling criteria, an example distribution and segmentation performance of each visual category can be seen in Table 1. Note that although the output of the segmentation routine produces a binary image (2 classes: air hydrates and background), the introduction of the five hydrate categories in the GT image is useful for a deeper understanding of the routines performance as well as the raw image data.

**Table 1.** Air hydrate visual categories and their segmentation probability for the GT image of the sample from 1404 m depth.

| Visual catgeory        | Labeling criteria                                             | Figure 1 | Labeled objects | Segmented |
|------------------------|---------------------------------------------------------------|----------|-----------------|-----------|
| Above focus (AF)       | Bright rim, dark core, rim 4-6 px                             | Е        | 41              | 34 (83%)  |
| In focus               | Core brightness similar to ice background, rim $<4$ px        | A,C      | 36              | 34 (94%)  |
| Below focus (BF)       | Dark rim, bright core, rim 4-6 px                             | D        | 65              | 57 (88%)  |
| Far out of focus (FOF) | Blurred core and rim structures, rim >6 px                    | В        | 173             | 127 (73%) |
| (X)                    | Clearly identified only by comparison with other focus levels | F        | 42              | 11 (26%)  |
| Artifacts              | -                                                             | e.g. I-V | -               | 17 (6%)   |
| Total objects          | -                                                             | -        | 357             | 280 (78%) |

#### 2.4.1 Evaluation of individual air hydrates

To determine the amount of detected air hydrates for each category (c.f. Table 1), each air hydrate together with its by 12 pixel enlarged bounding box was cut out from the GT image and automatically compared with the corresponding cutout in the segmented image. The air hydrate was detected if pixels in the cutouts overlap, while an "empty" cutout of the segmented image, or the lack of overlapping pixels, implies that the air hydrate was not detected. We then calculated the percentage of under-segmentation, i.e. the proportion of unrecognized hydrates, and the percentage of over-segmentation, i.e. the proportion of erroneously segmented objects (c.f. Table 2). The data for all GT images are shown in Table B1 and were additionally verified by visual inspection.

To determine the size discrepancy between GT- and segmented air hydrates, we compared the air hydrates' sizes in the above mentioned cutouts. Figure 3 shows the stacked distribution of differences in diameter between the GT- and segmented air hydrates in %. We defined the measurement error of the air hydrate diameter as the standard deviation of this distribution considering the negative skew (-22% / +14%). Air hydrates which are far out of focus (e.g. B in Fig. 1) cause the negative skewed distribution and are the main component of this error (Fig. 3). Note that the manual tracing of air hydrates far out of focus should be considered as a "best estimate" due to the blurred core and rim structures. In conclusion, the true size will be preferentially underestimated.

Figure 3. Stacked distribution of diameter discrepancy for each air hydrate in the eight GT- compared to the corresponding segmented image. The different colors and abbreviations correspond to the different visual categories in Table 1. The mean value and standard deviation are -4% and  $\pm 18\%$ , respectively.

Table 2. Segmentation metrics for the eight GT images.

| GT sample          | Under-seg. | Over-seg. | Balanced accuracy | Precision | Recall | F1   |
|--------------------|------------|-----------|-------------------|-----------|--------|------|
| 1284 m             | 13 %       | 4 %       | 86 %              | 95 %      | 71 %   | 81 % |
| 1314 m             | 24 %       | 13 %      | 85 %              | 85 %      | 61 %   | 71 % |
| 1404 m             | 26 %       | 6 %       | 85 %              | 82 %      | 71 %   | 76 % |
| $1464\;\mathrm{m}$ | 14 %       | 7 %       | 89 %              | 92 %      | 79 %   | 85 % |
| $1566\;\mathrm{m}$ | 22 %       | 11 %      | 80 %              | 91 %      | 60 %   | 72 % |
| 1755 m             | 23 %       | 5 %       | 84 %              | 89 %      | 69 %   | 78 % |
| $2265\;\mathrm{m}$ | 11 %       | 11 %      | 88 %              | 87 %      | 76 %   | 81 % |
| 2763 m             | 19 %       | 16 %      | 82 %              | 92 %      | 65 %   | 76 % |

# 2.4.2 Evaluation of general segmentation quality

To evaluate the general segmentation quality, we compared the GT images pixel by pixel with the segmented images. Each pixel in the segmented image can adopt one of four properties: 1) True Positive (TP) - Pixel is correctly segmented as air hydrate, 2) False Positive (FP) - Pixel is incorrectly segmented as air hydrate, 3) True Negative (TN) - Pixel is correctly segmented as background and 4) False Negative (FN) - Pixel is incorrectly segmented as background. These values are then the basis for calculating specific metrics to determine the quality of the segmentation routine (see equations 4 - 7).

The accuracy score shows the proportion of pixels correctly segmented, among all the pixels in the image. This metric can be biased if there is an imbalance in classes, e.g. background pixels and air hydrate pixels. Therefore, we use the balanced accuracy score defined as the average accuracy obtained on either class (Brodersen et al., 2010).

$$Balanced\ accuracy = \frac{1}{2} \left( \frac{TP}{TP + FN} + \frac{TN}{TN + FP} \right) \cdot 100\% \tag{4}$$

The precision metric is the proportion of pixels actually labeled as air hydrates among all the pixels segmented as air hydrates.

$$Precision = \frac{TP}{TP + FP} \cdot 100\% \tag{5}$$

170 Recall is defined as the proportion of pixels correctly segmented as air hydrates among all the pixels that are labeled air hydrate pixels.

$$Recall = \frac{TP}{TP + FN} \cdot 100\% \tag{6}$$

The F1-score calculates the harmonic mean between precision and recall and balances the error contribution of false negatives and false positives.

175 
$$F1 = \frac{2TP}{2TP + FP + FN} \cdot 100\% \tag{7}$$

We determined the pixelwise segmentation metrics (see Table 2) to optimize the segmentation routine (section 2.3.2) and to allow for a possible performance comparison with future segmentation methods.

#### 2.5 Further considerations for data interpretation

## 2.5.1 Convert air hydrate counts to air hydrate number concentration

We convert the measured air hydrate counts per sample to  $N_{ah}$  using the measured ice-sample thickness (mean value of 6 consecutive measurements; typically  $4.53 \pm 0.49$  mm) and the cropped mosaic image dimensions (80 x 30 mm). Air hydrate counts per sample and  $N_{ah}$  have a good linear correlation (Fig. 4) and we conclude that the observed volume (i.e. hydrate-mapping depth; Fig. 1) is consistent for all samples. An exception could be the samples from 1255 - 1294 m (Star-markers in Fig. 4).

Figure 4. Air hydrate counts per sample plotted against number concentration (R = -0.94, p 

Overall, the depth profiles of air hydrate number concentration and mean diameter show a trend of decreasing  $N_{ah}$  and increasing  $\overline{D}$  with increasing depth and thus increasing age, and they practically co-vary with the  $\delta^{18}$ O signal (Fig. 5a,b,c).

# 3.1.1 Comparison of air hydrate number concentration with air bubble concentration at EDML

Assuming a 1:1 conversion ratio between air bubbles and air hydrates, we estimate the detection rate of our method by comparing our measured  $N_{ah}$  with  $N_{ab}$  at EDML. Bendel et al. (2013) report an air bubble number concentration ( $N_{ab}$ ) of 300 - 400 cm<sup>-3</sup> for Holocene-ice and 400 - 500 cm<sup>-3</sup> for ice formed during the Last Glacial Maximum (LGM). Note that LGM-ice at EDML is located within the BHTZ (at about 1000 m depth), therefore, air bubbles are already converting to air hydrates. Lipenkov (2018) describe a semi-empirical model relating air bubble number and sizes to climate parameters and ice formation conditions (firn temperature, accumulation rate and surface snow density). Using this model we estimate  $N_{ab}$  for present-day conditions at EDML (-44.5 °C; 6.4 g cm<sup>-2</sup> yr<sup>-1</sup>; 0.38 g cm<sup>-3</sup>) to be about 400 cm<sup>-3</sup> and  $N_{ab}$  in the LGM ice (assuming: -54 °C; 3 g cm<sup>-2</sup> yr<sup>-1</sup>; 0.38 g cm<sup>-3</sup>) to be about 475 cm<sup>-3</sup>. We then compare our measured  $N_{ab}$  for MIS 5e, as an analogue for Holocene

Figure 5. Air hydrate depth profiles of the EDML ice core. a) Paleo-climatic information as  $\delta^{18}O_{ice}$  values. Blue corresponds to the dated-(EPICA Community Members, 2010), gray to the non-dated part of the ice core (Meyer et al.). b) air hydrate counts and number concentration  $(N_{ah})$ . Error-band for air hydrate counts is explained in section 2.4 and Table 2, error-bars are calculated including the uncertainty in sample thickness measurements. c) mean diameter  $(\overline{D})$  and relative standard deviation (RSD) and d) ice grain size (Weikusat et al., 2009). The ice age at the top of the figure corresponds to the AICC 2023 for EDML (Bouchet et al., 2023). The MIS boundaries are defined by Lisiecki and Raymo (2005), the MIS 5 substage boundaries are from Otvos (2015). Shaded bands correspond to "region 1" and "region 2" discussed in the text (especially in sections 4.2.1 and 4.2.2).

Figure 6. a)  $N_{ah}$  plotted against  $\overline{D}$ . The linear correlation and the Pearson Correlation Coefficient are calculated without considering data from 1255 - 1294 m (star markers) and region 2 (R = -0.92, p < 0.001). b)  $\delta^{18}O_{ice}$  plotted against  $N_{ah}$ . The linear correlation and the Pearson Correlation Coefficient are calculated without considering samples from 1255 - 1294 m and deeper than 2415 m (R = -0.57, p < 0.001). Square markers correspond to samples from 2025 - 2115 m depth (region 1 in Fig. 5,8). Triangular markers correspond to samples from 2382 - 2545 m depth (region 2 in Fig. 5,8).

conditions, and MIS 6, to represent the LGM, with  $N_{ab}$ . As a result, we detect between 48% and 63% of the expected  $N_{ah}$ . One explanation could be that the hydrate-mapping depth in our microscopic set-up might not correspond to the physical sample thickness (Fig. 1). In other words, microphotographs from one focus plane might not display the entire sample volume.

## 3.2 Air hydrate mean volume, volume concentration and air content

230

240

250

Figure 7 shows the mean volume  $(V_m)$  and volume concentration  $(V_c)$  of air hydrates as well as the theoretical total air content included in the air hydrate crystals  $(TAC_h)$ . We estimated the average volume of an air hydrate  $(V_m)$  using the arithmetic mean of the volumes of all air hydrates included in a sample.  $V_m$  is increasing with depth from about  $1.3 \cdot 10^{-6}$  to about  $3.0 \cdot 10^{-6}$  cm<sup>3</sup> (Fig. 7a). The volume concentration is defined as:

$$V_c = V_m \cdot N_{ah} \tag{8}$$

and is shown in Figure 7b. It decreases from  $5.0 \cdot 10^{-4}$  at  $1255 \,\mathrm{m}$  to  $3.0 \cdot 10^{-4}$  at about  $1700 \,\mathrm{m}$ . Then,  $V_c$  stays relatively constant at about  $3.4 \cdot 10^{-4}$ . A pronounced exception is observed for samples in the depth region from  $2382 - 2545 \,\mathrm{m}$ , where the  $V_c$  increases to an average of  $5.7 \cdot 10^{-4}$ .

Using the air hydrates' crystallographic parameters, we estimated the  $TAC_h$  (Fig. 7b) at standard pressure and temperature following Lipenkov (2000):

$$TAC_{h} = \frac{n\alpha}{a^{3}N_{L}} \cdot \frac{V_{c}}{\rho} \tag{9}$$

In above equation, a is the air hydrate lattice constant  $(1.72 \cdot 10^{-7} \text{ cm})$ , n is the amount of cages per unit cell (24) and  $\alpha$  is the cage occupancy (0.9; Lipenkov, 2000; Takeya et al., 2000).  $N_L$  is the Loschmidt constant  $(2.69 \cdot 10^{19} \text{ cm}^{-3})$  and  $\rho$  corresponds to the density of pure ice (0.92 g cm<sup>-3</sup>). Naturally, the  $TAC_h$  and  $V_c$  signals show the same pattern (Fig. 7b). The  $TAC_h$  decreases from 0.085 cm<sup>3</sup> g<sup>-1</sup> at 1255 m to an average of 0.059 cm<sup>3</sup> g<sup>-1</sup> from 1700 m down to the bottom of the ice core. From 2382 - 2545 m, the average  $TAC_h$  is 0.098 cm<sup>3</sup> g<sup>-1</sup>.

# 3.2.1 Comparison of $TAC_h$ with total air content at EDML

The total air content (TAC) in polar ice is considered to depend on air pressure, temperature and the pore volume in the firm at the time of pore close-off (e.g. Martinerie et al., 1992). As air hydrates contain most of the ancient air molecules in polar ice (Uchida et al., 2011), we can compare the calculated  $TAC_h$  with other independent TAC estimates. For EDML, a mean TAC of 0.0815 cm<sup>3</sup> g<sup>-1</sup> was measured by a method integrated to a Continuous Flow Analysis system (Ruth et al., 2004). This average value is lower than the TAC measured with absolute methods in Holocene ice cores from higher elevated drilling sites in Antarctica (Vostok, EDC; Martinerie et al., 1992). Therefore, we compare the calculated  $TAC_h$  to the theoretical TAC expected at EDML for an elevation of 2892 m a.s.l. and an atmospheric pressure of 650 - 700 mbar (about 0.092 cm<sup>3</sup> g<sup>-1</sup>) following Martinerie et al. (1992). The average  $TAC_h$  from 1700 m down to the bottom is 0.059 cm<sup>3</sup> g<sup>-1</sup>. This is about 64% of the expected TAC and matches well with our estimations comparing  $N_{ab}$  and  $N_{ah}$ . Employing a scaling factor for the  $N_{ah}$  of 1.55, the average  $TAC_h$  for this depth region matches the expected TAC value (orange markers in Fig. 7b). Note that this scaling was not applied for the graphs and the data interpretation.

The high  $TAC_h$  values for the depth range from 1255 - 1700 m will be discussed in chapter 4.1.1. The high  $TAC_h$  values for the depth range from and 2382 - 2545 m (region 2) will be discussed in chapters 4.1.2 and 4.2.2.

**Figure 7.** a) Air hydrate mean volume. The solid line represents the calculated growth rate for the dated part of the EDML ice core (1255 - 2415 m; Ruth et al., 2007). b) Blue markers: air hydrate volume concentration and total air content calculated from theoretical air content of air hydrates. Orange markers: scaled values to match expected mean TAC for EDML (dotted line at  $0.092 \, \mathrm{cm}^3 \, \mathrm{g}^{-1}$ ; Martinerie et al., 1992). Star markers correspond to samples from 1255 - 1294 m depth. Square markers correspond to samples from 2025 - 2115 m depth (region 1 in Figs. 5,8). Triangular markers correspond to samples from 2382 - 2545 m depth (region 2 in Figs. 5,8). The ice age at the top of the figure corresponds to the AICC 2023 for EDML (Bouchet et al., 2023).

#### 3.3 Air hydrate shape characteristics

260

265

The results of the air hydrate shape analysis together with the  $\delta^{18}O_{ice}$  record, the measured borehole temperature (Weikusat et al., 2017) and the second eigenvalue of the orientation tensor of the ice crystals c-axis distribution (Weikusat et al., 2013) are presented in Figure 8. The air hydrates' median aspect-ratio decreases with depth, which means they become rounder (Fig. 8b). From 1255 - 2005 m, the AR is relatively constant with an average of 1.40, however, between 2005 m and 2035 m, we observe a distinctive increase. From 2035 - 2395 m the average AR is 1.50 with outliers at 2115 m, 2125 m, 2205 m and 2305 m. From 2405 m down to the bottom, the AR decreases to an average value of 1.30. For every sample in the dataset, 90 % of measured air hydrates have an AR between 2.6 and 1.7 or smaller (Fig. 8c).

The median of the distribution of absolute air hydrate orientations, ( $|\alpha|$ ), are shown in Figure 8b. The average orientation from

Figure 8. Shape characteristics of air hydrates in the EDML ice core. a) Paleo-climatic information as  $\delta^{18}O_{ice}$  values. Blue corresponds to the dated- (EPICA Community Members, 2010), gray to the non-dated part of the ice core (Meyer et al.), b) air hydrate median AR and median of the absolute orientations ( $|\alpha|$ ), c) 90% percentile of air hydrate AR and d) measured borehole temperature (Weikusat et al., 2017) and the second eigenvalue of the orientation tensor of the ice crystals c-axis distribution ( $\lambda_2$ ) measured on vertical sections (Weikusat et al., 2013). Schematic stereographic projections are added to display the respective changes in ice CPO indicated by the three "terraces" in  $\lambda_2$ . The complete CPO of EDML is described in Weikusat et al. (2017). The ice age at the top of the figure corresponds to the AICC 2023 for EDML (Bouchet et al., 2023). The MIS boundaries are defined by Lisiecki and Raymo (2005), the MIS 5 substage boundaries are from Otvos (2015). Shaded bands correspond to "region 1" and "region 2" discussed in the text (especially in sections 4.2.1 and 4.2.2).

1255 m to 2005 m is 19° with a small increase of ca. 5° at around 1695 m. Similarly to the air hydrates median AR, we observe an abrupt decrease of  $|\alpha|$  of about 10° between 2005 m and 2035 m depth, i.e. a rotation towards the horizontal. From 2035 m,  $|\alpha|$  slowly decreases to a minimum of 7° at 2395 m. Between 2395 m and 2405 m  $|\alpha|$  suddenly doubles (14°), and from 2405 m to the bottom, it slowly decreases to about 10°. Overall,  $|\alpha|$  decreases with increasing depth.

#### 270 4 Discussion

# 4.1 Climate signal and depth variations of $N_{ah}$ and $\overline{D}$ at EDML

The EDML ice core chronology ends at 2415 m depth (Ruth et al., 2007). Below this depth, due to large scale disturbances in the ice stratigraphy, the integrity of the  $\delta^{18}$ O record is lost (Ruth et al., 2007; Faria et al., 2010a). For this chapter, we divide the record into a dated- (1255 - 2415 m) and not-dated (2415 - 2771 m) section.

# 275 4.1.1 Climate signal and depth variations of $N_{ah}$ and $\overline{D}$ from 1255 - 2415 m

Consistent with previous reports (e.g., Pauer et al., 1999; Lipenkov, 2000),  $N_{ah}$  and mean size point to a paleo-climatic influence on the air hydrate ensemble (Fig. 6a). We observe a higher  $N_{ah}$  during cold periods (e.g. MIS 4 and MIS 6) compared to warmer periods (e.g. MIS3 and MIS 5e). This means that the MIS 4 cold period and the penultimate glacial to interglacial transition are well resolved in the  $N_{ah}$  record of EDML. The correlation between  $\delta^{18}$ O and  $N_{ah}$  is shown in Figure 6b, and in the depth region from 1314 m to 2415 m, we observe an overall trend of decreasing  $N_{ah}$  with increasing  $\delta^{18}$ O values. Similarly, the air hydrates' mean diameter shows a connection to past climate (Fig. 5a,c), which is a consequence of the good linear correlation with  $N_{ah}$  (Fig. 6a). The good correlation of  $N_{ah}$  and  $\delta^{18}$ O previously found for glacial terminations (e.g. Lipenkov, 2000; Ohno et al., 2004), however, is not always reflected in the smaller scale climate fluctuations investigated in EDML. Note that measured  $\delta^{18}$ O values are averages of 50 cm, while we investigate 10 cm pieces. This means that possible small-scale fluctuations of  $\delta^{18}$ O within the averaged 50 cm are not accounted for, which could explain some of the mismatch. On the other hand, the correlation between past accumulation and temperature (i.e. stable water isotopes of ice) on smaller time scales is not as obvious as in the case of global glacial-interglacial climate changes. Since the air hydrate number concentration and mean size depend on the temperature and accumulation rate prevailing during the snow to ice transformation (e.g. Spencer et al., 2006; Lipenkov, 2018), this may also be the reason for the weaker correlation between  $N_{ah}$  and  $\delta^{18}$ O. Furthermore, the difference in the ages of the ice and trapped air bubbles (Lipenkov and Salamatin, 2014), and therefore potentially air hydrates, could result in additional discrepancies. For samples from 2025 - 2115 m depth (region 1 in Fig. 5), encompassing MIS 5b and parts of MIS 5c, we observe an evident deviation of the climatic influence on the air hydrate ensemble. We measure a low  $N_{ah}$ and a relatively large mean diameter (c.f. section 4.2.1), whereas the colder climate during MIS 5b (Fig. 5a) should result in a relatively high  $N_{ah}$  and a small average size of air hydrates.

The overall increase in mean air hydrate volume (Fig. 7a) with depth indicates air hydrate growth, which is overprinted by the climatic induced fluctuations. From 1255 m to 2415 m, we estimate the average growth rate to be  $8.0 \cdot 10^{-12}$  cm<sup>3</sup> per year

(Fig. 7a). This is slightly higher than growth rates calculated for the Vostok  $(6.7 \cdot 10^{-12} \text{ cm}^3)$ ; Uchida et al., 1994a) and GRIP  $(5.1 \cdot 10^{-12} \text{ cm}^3)$ ; Pauer et al., 1999) ice cores. Air hydrate crystal growth is frequently explained by air molecule diffusion and Ostwald ripening (e.g., Salamatin et al., 2003; Uchida et al., 2011). Due to the greater surface curvature, smaller particles are more soluble than larger particles, which results in a concentration gradient towards larger particles and consequently to the growth of larger air hydrates at the expense of smaller ones.

From 1255 m to 1700 m (spanning about 26 kyr), the  $TAC_h$  decreases from 0.085 cm<sup>3</sup> g<sup>-1</sup> to 0.059 cm<sup>3</sup> g<sup>-1</sup> (Fig. 7b). For the same depth region, we observe a decrease in the RSD of the air hydrate  $\overline{D}$  distribution (Fig. 5c) and a strong decrease in the AR 90% percentile from 2.5 to 2 (Fig. 8c). One explanation for the overestimation of the  $TAC_h$  (relative to the average value of 0.059 cm<sup>3</sup> g<sup>-1</sup>) could be the fact that the air hydrates' true cage occupancy for this region is probably slightly lower than our estimated 0.9, as it depends on pressure and temperature (Chazallon and Kuhs, 2002). However, the variations in gas hydrate cage occupancies with different formation conditions (e.g. Uchida et al., 1999; Hachikubo et al., 2022) are not large enough to solely explain the observed deviation from the mean. Another reason could be an uncertainty of the air hydrate size, as the calculation of  $V_c$  is very sensitive to errors in size determination. In the depth region just below the BHTZ we observe more complex-shaped air hydrates (Fig. 9) compared to deeper parts, which is indicated in the data by the decrease in the RSD and the AR 90% percentile (Fig. 5, 8c). The complex-shaped air hydrates partly seem to show a recrystallization towards an equilibrium state (regular and rounded morphologies). For the Vostok ice core, Lipenkov (2000) reported a depth region of faster air hydrate crystal growth rates, extending about 300 m below the BHTZ (i.e. down to 1550 m), and explains this by the large number of small, oxygen-enriched air hydrates dissolving in this region. Down to the same depth, Suwa and Bender (2008) measured noisy  $\delta O_2/N_2$  ratios. Similarly, Oyabu et al. (2021) reported a large scatter in measured  $\delta O_2/N_2$  ratios for about 300 m (spanning 25 kyr) below the BHTZ for the Dome Fuji ice core. Therefore, we surmise that the region from about 1255 - 1700 m at EDML coincides with a region of relatively faster air hydrate growth and recrystallization and the air hydrate

**Figure 9.** Example of complex-shaped air hydrates. a) Cloud-like air hydrate with protrusion from 1285 m. The equivalent-circle diameter (D) is 286  $\mu$ m. b) Disc-like, flat? air hydrate from 1285 m (D = 189  $\mu$ m). c) Cloud-like, porous? air hydrate from 1355 m (D = 325  $\mu$ m). d) Disc-like, flat? air hydrate from 1355 m (D = 247  $\mu$ m). e) Hollow ring shaped air hydrate from 1405 m (D = 179  $\mu$ m). f) Cloud-like with protrusion from 1566 m (D = 202  $\mu$ m). Scale bars correspond to 100  $\mu$ m.

ensemble might not have reached an equilibrium state. In summary, the overestimation likely originates from uncertainties of measurements in air hydrate size, which increase with complex-shaped air hydrates in the images.

# 320 4.1.2 Climate signal and depth variations of $N_{ah}$ and $\overline{D}$ from 2415 - 2771 m

The average  $N_{ah}$  for samples from 2415 - 2496 m is 282 cm<sup>-3</sup>, which indicates that they belong to a cold period (probably MIS 6). This is in line with measured  $\delta^{18}$ O values (Meyer et al.) and small ice grain sizes (Weikusat et al., 2009) (Fig. 5a,d). The air hydrates' mean diameter for these samples, however, is evidently overestimated (triangular markers in Fig. 6a) and, therefore, deviates from the climate dependence. For samples at 2535 m and 2545 m, the  $N_{ah}$  decreases to 203 cm<sup>-3</sup> and 144 cm<sup>-3</sup>, respectively, which could indicate a climatic transition. From 2392 - 2545 m (region 2 in Figs. 5,8), we observe an increase of clusters containing two or more individual air hydrates, whose diameter is typically in the range of the samples  $\overline{D}$  (Fig. 10). We define clusters consisting out of two air hydrates as "simple" (Fig. 10a), and clusters consisting out of more than two air hydrates as "multi" (Fig. 10b,c). Multi-clusters are especially common in the depth region 2. The observation of these clusters explains the overestimation of the mean diameter, as our image analysis method cannot distinguish between "one" air hydrate and clustered "many" air hydrates. Furthermore, the emerging of the latter explains the high mean volume and the abnormal volume concentration and  $TAC_h$  (Fig. 7a,b), as the volume of the cluster is overestimated during calculation of the volume from an area assuming a spherical shape. Besides the overestimation in volume, the increase in air hydrate clusters also leads to an underestimation of  $N_{ah}$ . However, quantification is challenging due to the difficulty in distinguishing individual air hydrate crystals within clusters from the images. Another indicator for increasing air hydrate clustering is the jump in the RSD of the mean diameter distribution between 2385 and 2392 m depth (Fig. 5c).

Figure 10. Example of clustered air hydrates in the sample from 2405 m depth. Scale bars correspond to 100  $\mu$ m. a) represents a "simple" clustered air hydrate, b) and c) represent "multi" clustered air hydrates. The equivalent-circle diameter for a), b) and c) is 257, 299 and 296  $\mu$ m, respectively, which is about two times the samples  $\overline{D}$  (137  $\mu$ m).

Below about 2550 m, both the  $N_{ah}$  and mean size clearly deviate from the paleo-climatic influence (gray circles in Fig. 6b). On the one hand, this could be due to the increasing large scale disturbances affecting the integrity of the  $\delta^{18}$ O record at this depth (Ruth et al., 2007; Faria et al., 2010a). On the other hand, air hydrates in deep ice are known to be affected by crystal growth, which can alter the climate-dependent variations of  $N_{ah}$  and mean size (Salamatin et al., 2003; Uchida et al., 2011). Evidences for air hydrate growth at EDML from about 2550 m are the decrease of the  $N_{ah}$  to the lowest-, and simultaneously, the increase of the  $\overline{D}$  to the highest values of the entire record (Fig. 5). Furthermore,  $N_{ah}$  and  $\overline{D}$  show a low variability, the air hydrates' mean volume increases steadily (Fig. 7a) and the air hydrates' aspect-ratio decreases (i.e. they become increasingly spherical; Fig 8b,c).

Air hydrate growth and the subsequent disappearance of the relationship between  $\delta^{18}O$  and air hydrate properties was reported for ice deeper than 2500 m at Dome Fuji (i.e. older than 350 - 400 kyr; Uchida et al., 2011), deeper than about 3300 m at Vostok (i.e. older than 300 - 400 kyr; Lipenkov et al., 2019) and for about 250 m from the bottom of the GRIP ice core in Greenland (Pauer et al., 1999; Salamatin et al., 2003). The fundamental reason behind the disappearance of climate-related variations in  $N_{ah}$  and  $\overline{D}$  is the fact that air hydrate growth rates are inversely proportional to the cube of their mean initial size (Salamatin et al., 2003). In other words, the growth rates of air hydrate ensembles with smaller mean size are higher than growth rates for ensembles with a larger mean size. Other parameters that influence air hydrate growth are increasing depth (and pressure), increasing age and increasing ice temperature (Uchida et al., 2011).

For the EDML ice core, the age of the ice deeper than 2415 m is not known. However, the sudden increase in ice grain size by one to two magnitudes to up to 0.5 - 1 m for the deepest samples (Faria et al., 2010a; Weikusat et al., 2017) and the relatively warm in-situ ice temperature of -8°C at about 2550 m (Figure 8d) indicate favorable conditions for air hydrate crystal growth. Thus, we interpret our data to mean that from this depth, the climate signal is dampened by considerable air hydrate crystal growth.

The reason for the absence of air hydrates in the lowermost sample at a depth of 2773.91 m, that is, about 25 cm above the bottom of the ice core, is currently unclear. The bottom of the core reaches the pressure melting point (-2°C; Faria et al., 2010a) and ice-core drilling was terminated because subglacial water entered the borehole (Wilhelms et al., 2014). Furthermore, Faria et al. (2010a) reported that, from 2760 m, the concentration of micro inclusions gradually decreases with depth. Therefore, the sample could represent accreted ice with no or very little impurity inclusions (at least in the 10 cm sample!). On the other hand, the ice could be of meteoric origin and subject to processes that resulted in the expulsion of impurities and gases, i.e. the segregation of impurities to the grain boundary network and subsequent drainage to the bedrock driven by the hydraulic gradient (Rempel et al., 2002; Rempel, 2005). These processes are considered to alter the basal ice at EDC (Tison et al., 2015), Dome Fuji (Ohno et al., 2016) and NEEM (Goossens et al., 2016). Note that a decrease in TAC was only reported (or measured) for samples of the NEEM ice core (Goossens et al., 2016). Further investigations of the deepest ice at EDML are necessary to determine its origin and explain the absence of air hydrates.

**Table 3.** Air hydrate parameters for selected samples from MIS 3-5 and from the penultimate glacial / interglacial transition zone. We used randomly selected air hydrates belonging to the "in focus" visual category (A, C in Fig. 1) to determine the amount of clustered air hydrates. The amount of randomly selected air hydrates is equal to the samples'  $N_{ah}$ .

| Sample             | MIS | $N_{ah}$ | mean diameter [ $\mu$ m] | median $ \alpha $ | median AR | clustered  |
|--------------------|-----|----------|--------------------------|-------------------|-----------|------------|
| 1635 m             | 3   | 266      | 120                      | 17                | 1.42      | 8%         |
| $1755\;\mathrm{m}$ | 4   | 320      | 115                      | 20                | 1.41      | 10%        |
| $2035\;\mathrm{m}$ | 5b  | 206      | 141                      | 10                | 1.47      | 7%         |
| $2375\;\mathrm{m}$ | 5e  | 194      | 139                      | 7                 | 1.48      | 7%         |
| $2382\;\mathrm{m}$ | 5e  | 234      | 147                      | 7                 | 1.51      | 6%         |
| $2385\;\mathrm{m}$ | 6   | 192      | 142                      | 8                 | 1.53      | 8%         |
| $2392\;\mathrm{m}$ | 6   | 271      | 139                      | 7                 | 1.48      | 21%, multi |
| $2395\;\mathrm{m}$ | 6   | 270      | 139                      | 7                 | 1.48      | 19%, multi |
| $2405\;\mathrm{m}$ | 6   | 292      | 137                      | 14                | 1.35      | 23%, multi |
| $2425\;\mathrm{m}$ | -   | 270      | 141                      | 12                | 1.28      | 20%, multi |
| 2652 m             | -   | 124      | 150                      | 11                | 1.35      | 5%         |

# 4.2 Influence of ice deformation and microstructure on the air hydrate ensemble

### 4.2.1 Region 1; 2025 - 2115 m

For this depth range, corresponding to region 1, we observe an inverse behavior of  $N_{ah}$  and  $\overline{D}$  to what would be expected based on the  $\delta^{18}$ O record (Fig. 5a,b,c). On top of that, we observe a sudden increase in air hydrate aspect-ratio and a rotation of their major axis towards the horizontal between 2005 and 2035 m, i.e. at the beginning of the mismatching region.

At approximately 2030 m depth, Weikusat et al. (2017) observed an abrupt change in the ice crystallographic preferred orientation (CPO) from a vertical girdle to a single maximum (indicated in our Fig. 8d), and attribute this observation to a transition from a pure shear to a simple shear dominated deformation regime. Additionally, inclined and undulating cloudy bands on the mm- to cm-scale (i.e. a disturbed stratigraphy) and the onset of an echo-free zone in radio-echo sounding measurements were observed from about 2050 m depth (Drews et al., 2009; Faria et al., 2010a, b). We surmise that the simultaneous occurrence of the abrupt change in ice crystal properties, as a consequence of change in overall deformation regime, and the change in air hydrate shape parameters are linked.

Durham et al. (2010) summarized the rheological behavior of certain sI and sII gas hydrates measured in laboratory deformation tests and found them to be several orders of magnitude stronger than ice. Cao et al. (2020) performed molecular simulations to assess the mechanical behavior of ice-contained methane hydrates (sI) and found their partial dissociation and reformation due to mechanical deformation. Therefore, we suspect that air hydrates in ice could be seen as an analog to porphyroclasts in rocks, which are single (mineral-) grains, that have a different rheology than the surrounding matrix. Porphyroclasts may be rigid, deformable or weaker than their surrounding matrix and have the ability to recrystallize, rotate or elongate and obtain a

stable position in the extensional quadrant of flow (e.g., Passchier and Trouw, 2005). The increase in air hydrate aspect ratio and their rotation towards the image x-axis (shear plane?) from about 2035 m could be caused by crystal-plastic deformation combined with rigid body rotation and matches well with the change towards a simple shear dominated deformation regime, considering the aforementioned behavior of porphyroclasts in rocks.

An additional explanation could be that air hydrates preferentially elongate along the basal plane of ice crystals. Using molecular dynamics simulations, Factorovich et al. (2019) found that the (111) lattice plane of the sII hydrate can bind to the basal plane of hexagonal ice through domain matching to produce epitaxy and alignment between these crystals. Similarly, a strong connection of the ice crystal basal plane and air bubble elongation was found by Fegyveresi et al. (2019). Considering the CPO change to a single maximum, a preferred elongation of air hydrates along the basal plane of ice crystals could explain the sudden rotation towards the image x-axis between 2005 m and 2035 m.

The air hydrates' elongation and alignment towards the horizontal persists down to 2395 m depth and could be explained by the above mentioned mechanisms. The strong anomaly in  $N_{ah}$  and  $\overline{D}$ , however, is restricted to 2025 - 2115 m. Wether the change in deformation regime can cause a localized air hydrate growth due to, for instance, a change in the air molecule diffusion rate caused by an increase in interstitials and lattice defects in the ice matrix, requires further investigations.

# 400 **4.2.2 Region 2; 2392 - 2545** m

For this depth range, corresponding to region 2, we observe an increase in air hydrate clusters, which cause an overestimation of the mean diameter (Figs. 5,6a and Table 3). On the one hand, the air hydrate clusters could form from fragmentation of larger, "out-of-equilibrium" air hydrates. On the other hand, they could form due to coalescence, which occurs when encountering air hydrates merge in order to reduce their total interfacial area (Uchida et al., 2011). We suspect the latter: Firstly, the depth region from 2385 - 2495 m corresponds to a cold climate and, as a result, should contain (and does contain) a higher amount of relatively smaller air hydrates. Secondly, the diameter of the individual air hydrates forming the clusters is in the range of the samples  $\overline{D}$  (Fig. 10) and, thirdly, the majority of air hydrates are rounded and have a low aspect-ratio (Fig. 8b,c). As a prerequisite for coalescence, air hydrate crystals need to migrate through the ice matrix, and one underlying mechanism could be dragging due to grain boundary migration during ice crystal growth (Uchida et al., 2011). The ice core depth region from 2392 - 2545 m seems predestined for such a process. The appearance of a distinctive ice microstructure ("brick wall pattern") is evidence for strain accommodation via grain boundary sliding (Faria et al., 2009). Moreover, from 2405 m to about 2600 m, cloudy bands are inclined, and folding as well as other stratigraphic disturbances are frequent (Faria et al., 2010a, b). These observations are indicators for high strain rates and a bed-parallel (simple) shear dominated deformation regime (Faria et al., 2010a; Weikusat et al., 2017).

#### 415 5 Conclusions

405

We present a new method to analyze air hydrate crystals in high resolution microphotographs of polished ice thick sections using automated image segmentation and analysis. This new image analysis approach allows an efficient evaluation of large

amounts of image and air hydrate data and eliminates the need for tedious manual counting. We analyzed air hydrates included in 74 samples of the EDML ice core over a depth range from 1255 m - 2771 m, which were acquired in the field (years 2003-2006) shortly after drilling the ice core. We measured the number of air hydrates in each sample and characterized their sizes 420 and shapes to obtain depth profiles of the number concentration  $(N_{ab})$ , mean diameter  $(\overline{D})$ , volume concentration  $(V_c)$ , aspectratio (AR) and orientation ( $\alpha$ ). The measured variations of  $N_{ah}$  and mean diameter with depth clearly resolve the MIS 4 stadial period and the penultimate glacial to interglacial transition. Consequently, we confirm the general relation of the air hydrate properties with past climate, as shown previously for the Dome Fuji (e.g. Ohno et al., 2004), Vostok (e.g. Lipenkov, 2000) and GRIP (Pauer et al., 1999) ice cores. We also confirm that air hydrates grow with depth (time) based on the observed decrease of  $N_{ah}$  and corresponding increase in mean size (i.e. mean volume) following an Ostwald ripening process. We estimate the growth rate from 1255 - 2405 m, which corresponds to the dated depth region of the EDML ice core, to be  $8.0 \cdot 10^{-12}$  cm<sup>3</sup> per year. Besides Ostwald ripening, locally increased air hydrate coalescence causes a minor decrease in  $N_{ab}$  and an increase in mean diameter in the depth range of 2392 m to 2545 m. From about 2550 m down to the bottom, the climatic control on the air hydrate ensemble is disturbed by air hydrate growth, which coincides with observations from the GRIP ice core (Pauer et al., 430 1999; Salamatin et al., 2003).

The connection between the  $\delta^{18}$ O and air hydrate records is not always resolved on the small scale at EDML. To pinpoint the reasons, detailed studies on e.g. the mechanisms for the air bubble to air hydrate transition are required. They should be carried out as soon as possible after drilling the ice core (Pauer et al., 1999; Kipfstuhl et al., 2001).

The EDML ice core was drilled on an ice divide and has a distinctive ice microstructure interpreted as the effects of changing boundary conditions of deformation (Weikusat et al., 2017). Remarkably, our air hydrate shape profiles, expressed as variations in median AR and median |α|, reflect the change from a pure shear to a simple shear dominated deformation regime at about 2030 m as discussed by Weikusat et al. (2017). We observe a sudden increase in air hydrates' AR as well as the rotation of their major axis towards the horizontal. Furthermore, increased air hydrate coalescence from 2392 - 2545 m coincides with a bed parallel shear dominated region exposed to high strain rates (Faria et al., 2010a; Weikusat et al., 2017). For this reason, we are convinced that the air hydrate ensemble is substantially influenced by ice deformation and / or the resulting ice microstructure. Identifying the mechanisms behind the interaction of air hydrates and ice (deformation) and whether these mechanisms can also cause the pronounced deviation of the climatic influence observed from 2025 - 2115 m depth should be investigated in future studies.

The scientific results, methods and workflows to analyze air hydrates in microphotographs described in this work will assist to improve our understanding of air hydrates and their properties in ice cores and can be useful to further develop a novel dating tool based on air hydrate crystal growth (Lipenkov et al., 2019).

Code and data availability. Code and data will be made publicly available via open-access repositories such as PANGAEA once the manuscript has been accepted. The  $\delta^{18}O_{ice}$  values for EDML from 2415 m to the bottom will be published in PANGAEA by Meyer et al..

**Figure A1.** Selected samples and their median gray value (i.e. brightness) acquired during three Antarctic field seasons in the depth range from 1255 m to 2771 m. The different colors correspond to the six image categories and the black bars mark the samples where a section of the sample was used as ground truth image.

# Appendix B: Image analysis

#### **B1** Air hydrate segmentation

The Canny edge detection algorithm (Canny, 1986) is the core of this image segmentation routine. It is a multi-stage edge detector and is implemented in scikit-image (van der Walt et al., 2014) in the following way: In a first step, the image is smoothed by a Gaussian filter to reduce noise. Next, the intensity gradients of the image are calculated, of which only the maxima are identified as a potential edges. At last, a hysteresis thresholding on the gradient magnitude is applied. Edges with an intensity gradient above a maximum threshold are true-edges and those below a minimum threshold are non-edges. Those who lie between these two thresholds are classified true-edges or non-edges based on their connectivity. If they are connected to "true-edge" pixels, they are considered to be part of edges. The prediction of true-edges via hysteresis thresholding depends a.o. on the image brightness. To account for the different imaging conditions, we kept the minimum threshold at zero, while the optimal maximum threshold was determined for each of the six image categories by optimizing the segmentation metrics for ground truth (GT) images. For this, we prioritized precision over recall to avoid over-segmentation (see section 2.4). The Canny edge detection operation produces a binary edge map and an example result is shown in Figure 2b. In a first segmentation step, a morphological closing (1 px as structuring element), a "fillholes" and an erosion (by 1 px) operation are applied to the edge-map, to get a first binary image of well-segmented objects (Segmented Mask 1 = SM 1 in Fig. B1a, red objects in Fig. B1c). Next, SM1 is used as a mask for another edge detection operation in second segmentation step. From the resulting edge-map, objects with a small area and a high aspect-ratio (usually erroneously segmented grain boundaries) are removed before a "force-closing" (disk with r=3 px as structuring element), a "fillholes" and an erosion operation are applied to produce a second mask (SM 2 in Fig. B1a, blue objects in Fig. B1c) Ultimately, a set of filters are applied to the resulting SM 1 and SM 2 images to remove common artifacts (e.g. I-V in Fig. 1) before combining them to the final segmentation result (Fig. 2c).

**Figure B1.** Detailed segmentation (a) and filtering (b) steps. c) shows SM 1 and SM 2 after the segmentation step (a) on top of the original image.

# **B2** Segmentation evaluation with ground truth images

**Table B1.** Comparing ground truth and segmentation results. The table headers correspond to air hydrate visual categories (see Table 1).

| GT sample |           | AF        | In focus  | BF        | FOF       | X        | Artifacts | Total     |
|-----------|-----------|-----------|-----------|-----------|-----------|----------|-----------|-----------|
| 1284 m    | labeled   | 54        | 35        | 102       | 137       | 31       | -         | 359       |
|           | segmented | 53 (98%)  | 30 (86%)  | 98 (96%)  | 114 (83%) | 18 (58%) | 12 (4%)   | 325 (91%) |
| 1314 m    | labeled   | 37        | 58        | 73        | 128       | 61       | -         | 357       |
|           | segmented | 32 (86%)  | 53 (91%)  | 70 (96%)  | 90 (70%)  | 25 (41%) | 41 (13%)  | 311 (87%) |
| 1404 m    | labeled   | 41        | 36        | 65        | 173       | 42       | -         | 357       |
|           | segmented | 34 (83%)  | 34 (94%)  | 57 (88%)  | 127 (73%) | 11 (26%) | 17 (6%)   | 280 (78%) |
| 1464 m    | labeled   | 27        | 29        | 76        | 128       | 18       | -         | 278       |
|           | segmented | 25 (93%)  | 28 (97%)  | 74 (97%)  | 105 (82%) | 8 (44%)  | 18 (7%)   | 258 (93%) |
| 1566 m    | labeled   | 32        | 28        | 38        | 134       | 24       | -         | 256       |
|           | segmented | 28 (88)   | 27 (96%)  | 38 (100%) | 97 (72%)  | 10 (42%) | 25 (11%)  | 225 (88%) |
| 1755 m    | labeled   | 55        | 34        | 94        | 150       | 54       | -         | 387       |
|           | segmented | 50 (91%)  | 30 (88%)  | 90 (96%)  | 114 (76%) | 15 (28%) | 15 (5%)   | 314 (81%) |
| 2265 m    | labeled   | 22        | 27        | 44        | 79        | 12       | -         | 184       |
|           | segmented | 20 (83%)  | 26 (94%)  | 44 (88%)  | 71 (73%)  | 3 (26%)  | 17 (11%)  | 181 (98%) |
| 2763 m    | labeled   | 29        | 19        | 21        | 75        | 16       | -         | 160       |
|           | segmented | 29 (100%) | 19 (100%) | 21 (100%) | 60 (80%)  | 1 (6%)   | 24 (16%)  | 154 (96%) |

# **Appendix C: Projection properties of Ellipsoids**

Figures C1, C2, C3 show the changes in orthogonal projection properties of an ellipsoid that is rotated around the core axis (Z). A and C of the ellipsoid correspond to the major- and minor axis of the projected ellipse and are the same for the three initial conditions (Fig. C1a, C2a, C3a). The third dimension, B, is changed accordingly to represent three end-member cases.

**Figure C1.** a) shows an ellipsoid together with its projection to the XZ-plane, which corresponds to the image plane of the microphotograph (c.f. Fig.1). A=14 (red), B=10 (green), C=10 (blue) are the ellipsoids' axis and the orientation is the angle (-17°) between the semi-major axis (A) and the X-axis. b), c), d), e) show the evolution of the aspect-ratio, orientation, major- and minor axis and the relative area with progressing rotation around the Z-axis (core axis) from  $0^{\circ}$ , initial condition in panel (a), to  $360^{\circ}$ .

Figure C2. a) shows an ellipsoid together with its projection to the XZ-plane, which corresponds to the image plane of the microphotograph (c.f. Fig.1). A=14 (red), B=5 (green), C=10 (blue) are the ellipsoids' axis and the orientation is the angle (-17 $^{\circ}$ ) between the semi-major axis (A) and the X-axis. b), c), d), e) show the evolution of the aspect-ratio, orientation, major- and minor axis and the relative area with progressing rotation around the Z-axis (core axis) from  $0^{\circ}$ , initial condition in panel (a), to  $360^{\circ}$ .

**Figure C3.** a) shows an ellipsoid together with its projection to the XZ-plane, which corresponds to the image plane of the microphotograph (c.f. Fig.1). A=14 (red), B=15 (green), C=10 (blue) are the ellipsoids' axis and the orientation is the angle (-17°) between the semi-major axis (A) and the X-axis. b), c), d), e) show the evolution of the aspect-ratio, orientation, major- and minor axis and the relative area with progressing rotation around the Z-axis (core axis) from 0°, initial condition in panel (a), to 360°.

One challenge in the interpretation of shape and orientation data lies in the fact that we do not know if we measured the maximum elongation and that the third dimension is unknown, i.e. if the non-isometric object is more cigar (prolate)-like (e.g. Fig. C2) or more oblate-like (e.g. Fig. C3). For the interpretation of the air hydrate shape characteristics, we consider the relative orientation of the ice samples to each other. As mentioned in section 2.5.2, a change in relative orientation could be caused due to a difficulty in matching core breakpoints during ice-core logging.

Weikusat et al. (2017) report a loss in azimuthal orientation of the core between 1686 m and 1696 m as well as between 1955 m and 2035 m depth. Between 1665 m and 1695 m, we observe an increase in air hydrate median orientation from 17° to 22° (Fig. 8b) as well as a "polarity" change (Fig. C4a,b). However, we do not observe a significant change in the median AR (Fig. 8b). Between 2005 m and 2035 m depth, we observe a more continuous decrease in the median orientation (Fig. 8b and Fig. C4c,d,e) and an increase in the median AR (Fig. 8b). The fact that the median AR does not change between 1665 m and 1695 m, while the orientation does, implies that either the air hydrates are not very well sorted or that the majority is rather isometric in shape. On the contrary, the continuous and joint change of AR together with orientation from 2005 m to 2035 m depth point either towards an increase in sorting or a decrease in isometric air hydrates. Both could be induced by the increase in simple shear deformation.

Future work should consider taking images from two (or more) different directions to increase the accuracy in air hydrate shape and orientation measurements.

**Figure C4.** This figure shows orientations plots ("rose plots") of the air hydrate major axis for samples from a) 1665 m, b) 1695 m, c) 2005 m, d) 2025 m and e) 2035 m depth. "n" corresponds to the number of measured objects for each sample, i.e. objects with an AR higher than 1.33.

Author contributions. The study was conceptualized by FP, SK and IW. The microphotograph raw-data were collected by SK. The image and data analysis routines were developed by FP with support from IW and JF. An initial draft was produced by FP and all authors contributed to the discussion of the results and the final version of the manuscript.

Competing interests. The authors declare that they have no conflict of interest.

Acknowledgements. We thank Jan Eichler for providing the image stitching script. This publication was generated in the frame of DEEPICE project. The project has received funding from the European Union's Horizon 2020 research and innovation programme under the Marie Sklodowska-Curie grant agreement No 955750. The opinions expressed and arguments employed herein do not necessarily reflect the official views of the European Union funding agency or other national funding bodies. This work is a contribution to the European Project for Ice Coring in Antarctica (EPICA), a joint European Science Foundation/European Commission scientific programme, funded by the European Union (EPICA-MIS) and by national contributions from Belgium, Denmark, France, Germany, Italy, the Netherlands, Norway, Sweden, Switzerland and the United Kingdom. The main logistic support was provided AWI at Dronning Maud Land. This work is EPICA publication no. XXX.

- Arzt, M., Deschamps, J., Schmied, C., Pietzsch, T., Schmidt, D., Tomancak, P., Haase, R., and Jug, F.: LABKIT: Labeling and Segmentation Toolkit for Big Image Data, Frontiers in Computer Science, 4, https://doi.org/10.3389/fcomp.2022.777728, 2022.
- Bendel, V., Ueltzhöffer, K. J., Freitag, J., Kipfstuhl, S., Kuhs, W. F., Garbe, C. S., and Faria, S. H.: High-resolution variations in size, number and arrangement of air bubbles in the EPICA DML (Antarctica) ice core, Journal of Glaciology, 59, 972–980, https://doi.org/10.3189/2013JoG12J245, 2013.
- Binder, T., Garbe, C., Wagenbach, D., Freitag, J., and Kipfstuhl, S.: Extraction and parametrization of grain boundary networks in glacier ice, using a dedicated method of automatic image analysis, Journal of Microscopy, 250, 130–141, https://doi.org/10.1111/jmi.12029, 2013.
- Bouchet, M., Landais, A., Grisart, A., Parrenin, F., Prié, F., Jacob, R., Fourré, E., Capron, E., Raynaud, D., Lipenkov, V. Y., Loutre, M.-F., Extier, T., Svensson, A. M., Martinerie, P., Leuenberger, M. C., Jiang, W., Ritterbusch, F., Lu, Z.-T., and Yang, G.-M.: AICC2023, EDML,
- in: AICC2023, PANGAEA, https://doi.org/10.1594/PANGAEA.961019, in: Bouchet, M et al. (2023): The Antarctic ice core chronology (AICC2023) [dataset bundled publication]. PANGAEA, https://doi.org/10.1594/PANGAEA.961017, 2023.
  - Brodersen, K. H., Ong, C. S., Stephan, K. E., and Buhmann, J. M.: The Balanced Accuracy and Its Posterior Distribution, in: 2010 20th International Conference on Pattern Recognition, pp. 3121–3124, https://doi.org/10.1109/ICPR.2010.764, 2010.
  - Canny, J.: A Computational Approach to Edge Detection, IEEE Transactions on Pattern Analysis and Machine Intelligence, PAMI-8, 679–698, https://doi.org/10.1109/TPAMI.1986.4767851, 1986.
    - Cao, P., Ning, F., Wu, J., Cao, B., Li, T., Sveinsson, H. A., Liu, Z., Vlugt, T. J. H., and Hyodo, M.: Mechanical Response of Nanocrystalline Ice-Contained Methane Hydrates: Key Role of Water Ice, ACS Applied Materials & Interfaces, 12, 14016–14028, https://doi.org/10.1021/acsami.0c00972, 2020.
- Chazallon, B. and Kuhs, W. F.: In situ structural properties of N2-, O2-, and air-clathrates by neutron diffraction, The Journal of Chemical Physics, 117, 308–320, https://doi.org/10.1063/1.1480861, 2002.
  - Drews, R., Eisen, O., Weikusat, I., Kipfstuhl, S., Lambrecht, A., Steinhage, D., Wilhelms, F., and Miller, H.: Layer disturbances and the radio-echo free zone in ice sheets, The Cryosphere, 3, 195–203, https://doi.org/10.5194/tc-3-195-2009, 2009.
  - Durham, W. B., Prieto-Ballesteros, O., Goldsby, D. L., and Kargel, J. S.: Rheological and thermal properties of icy materials, Space Science Reviews, 153, 273–298, https://doi.org/10.1007/s11214-009-9619-1, 2010.
- 530 EPICA Community Members: One-to-one coupling of glacial climate variability in Greenland and Antarctica, Nature, 444, 195–198, https://doi.org/10.1038/nature05301, 2006.
  - EPICA Community Members: Stable oxygen isotopes of ice core EDML, https://doi.org/10.1594/PANGAEA.754444, 2010.
  - Factorovich, M. H., Naullage, P. M., and Molinero, V.: Can clathrates heterogeneously nucleate ice?, The Journal of Chemical Physics, 151, 114707, https://doi.org/10.1063/1.5119823, 2019.
- Faria, S. H., Kipfstuhl, S., Azuma, N., Freitag, J., Weikusat, I., Murshed, M. M., and Kuhs, W. F.: The multiscale structure of Antarctica. Part 1: inland ice, Institute of Low Temperature Science, Hokkaido University Sapporo, 68, 39–59, 2009.
  - Faria, S. H., Freitag, J., and Kipfstuhl, S.: Polar ice structure and the integrity of ice-core paleoclimate records, Quaternary Science Reviews, 29, 338–351, https://doi.org/10.1016/j.quascirev.2009.10.016, climate of the Last Million Years: New Insights from EPICA and Other Records, 2010a.
- Faria, S. H., Kipfstuhl, S., and Lambrecht, A.: The EPICA-DML Deep Ice Core: A Visual Record, Frontiers in Earth Sciences, Springer Berlin, Heidelberg, https://doi.org/10.1007/978-3-662-55308-4, 2010b.

- Fegyveresi, J. M., Alley, R. B., Voigt, D. E., Fitzpatrick, J. J., and Wilen, L. A.: Instruments and methods: a case study of ice core bubbles as strain indicators, Annals of Glaciology, 60, 8–19, https://doi.org/10.1017/aog.2018.23, 2019.
- Goossens, T., Sapart, C. J., Dahl-Jensen, D., Popp, T., El Amri, S., and Tison, J.-L.: A comprehensive interpretation of the NEEM basal ice build-up using a multi-parametric approach, The Cryosphere, 10, 553–567, https://doi.org/10.5194/tc-10-553-2016, 2016.
  - Hachikubo, A., Fuseya, G., Sugimori, E., and Takeya, S.: Effect of Pressure on the Hydration Number of Argon Hydrate, Journal of Chemical & Engineering Data, 67, 3757–3762, https://doi.org/10.1021/acs.jced.2c00602, 2022.
  - Hondoh, T., Anzai, H., Goto, A., Mae, S., Higashi, A., and Langway, C. C.: The crystallographic structure of the natural airhydrate in Greenland dye-3 deep ice core, Journal of inclusion phenomena and molecular recognition in chemistry, 8, 17–24, https://doi.org/10.1007/BF01131284, 1990.

550

- Hvidberg, C. S., Steffensen, J. P., Clausen, H. B., Shoji, H., and Kipfstuhl, J.: The NorthGRIP ice-core logging procedure: description and evaluation, Annals of Glaciology, 35, 5–8, https://doi.org/10.3189/172756402781817293, 2002.
- Jing, J., Liu, S., Wang, G., Zhang, W., and Sun, C.: Recent advances on image edge detection: A comprehensive review, Neurocomputing, 503, 259–271, https://doi.org/10.1016/j.neucom.2022.06.083, 2022.
- Kipfstuhl, S.: Thick-section images of the EPICA-Dronning-Maud-Land (EDML) ice core., https://doi.org/10.1594/PANGAEA.663141, 2007.
  - Kipfstuhl, S., Pauer, F., Kuhs, W. F., and Shoji, H.: Air bubbles and Clathrate hydrates in the transition zone of the NGRIP Deep Ice Core, Geophysical Research Letters, 28, 591–594, https://doi.org/10.1029/1999GL006094, 2001.
- Kipfstuhl, S., Hamann, I., Lambrecht, A., Freitag, J., Faria, S. H., Grigoriev, D., and Azuma, N.: Microstructure mapping: a new method for imaging deformation-induced microstructural features of ice on the grain scale, Journal of Glaciology, 52, 398–406, https://doi.org/10.3189/172756506781828647, 2006.
  - Kipfstuhl, S., Faria, S. H., Azuma, N., Freitag, J., Hamann, I., Kaufmann, P., Miller, H., Weiler, K., and Wilhelms, F.: Evidence of dynamic recrystallization in polar firm, Journal of Geophysical Research: Solid Earth, 114, https://doi.org/10.1029/2008JB005583, 2009.
  - Lipenkov, V. Y.: Air bubbles and air-hydrate crystals in the Vostok ice core, in: Physics of Ice Core Records, pp. 327–358, Hokkaido University Press, http://hdl.handle.net/2115/32474, 2000.
  - Lipenkov, V. Y.: How air bubbles form in polar ice, Earth's Cryosphere, 22, 16–28, https://www.researchgate.net/publication/326253969\_ How air bubbles form in polar ice, 2018.
  - Lipenkov, V. Y. and Salamatin, A. N.: Steady-state size distribution of air bubbles in polar ice, Ice and Snow, 54, 20–31, https://doi.org/10.15356/2076-6734-2014-4-20-31, 2014.
- Lipenkov, V. Y., Salamatin, A. N., Jiang, W., Ritterbusch, F., Bender, M. L., Orsi, A., Landais, A., Uchida, T., Ekaykin, A. A., Raynaud, D., et al.: New ice dating tools reveal 1.2 Ma old meteoric ice near the base of the Vostok ice core., in: Geophysical Research Abstracts, vol. 21, https://meetingorganizer.copernicus.org/EGU2019/EGU2019-8505.pdf, 2019.
  - Lisiecki, L. E. and Raymo, M. E.: A Pliocene-Pleistocene stack of 57 globally distributed benthic  $\delta$ 18O records, Paleoceanography, 20, https://doi.org/10.1029/2004PA001071, 2005.
- Martinerie, P., Raynaud, D., Etheridge, D. M., Barnola, J.-M., and Mazaudier, D.: Physical and climatic parameters which influence the air content in polar ice, Earth and Planetary Science Letters, 112, 1–13, https://doi.org/10.1016/0012-821X(92)90002-D, 1992.
  Meyer, H. et al.: To be published in PANGAEA.

Narita, H., Azuma, N., Hondoh, T., Fujii, M., Kawaguchi, M., Mae, S., Shoji, H., Kameda, T., and Watanabe, O.: Characteristics of air bubbles and hydrates in the Dome Fuji ice core, Antarctica, Annals of Glaciology, 29, 207–210, https://doi.org/10.3189/172756499781821300, 1999.

580

585

- Ohno, H., Lipenkov, V. Y., and Hondoh, T.: Air bubble to clathrate hydrate transformation in polar ice sheets: A reconsideration based on the new data from Dome Fuji ice core, Geophysical Research Letters, 31, https://doi.org/10.1029/2004GL021151, 2004.
- Ohno, H., Iizuka, Y., Hori, A., Miyamoto, A., Hirabayashi, M., Miyake, T., Kuramoto, T., Fujita, S., Segawa, T., Uemura, R., Sakurai, T., Suzuki, T., and Motoyama, H.: Physicochemical properties of bottom ice from Dome Fuji, inland East Antarctica, Journal of Geophysical Research: Earth Surface, 121, 1230–1250, https://doi.org/10.1002/2015JF003777, 2016.
- Otvos, E. G.: The Last Interglacial Stage: Definitions and marine highstand, North America and Eurasia, Quaternary International, 383, 158–173, https://doi.org/10.1016/j.quaint.2014.05.010, 2015.
- Oyabu, I., Kawamura, K., Uchida, T., Fujita, S., Kitamura, K., Hirabayashi, M., Aoki, S., Morimoto, S., Nakazawa, T., Severinghaus, J. P., and Morgan, J. D.: Fractionation of  $O_2/N_2$  and  $Ar/N_2$  in the Antarctic ice sheet during bubble formation and bubble–clathrate hydrate transition from precise gas measurements of the Dome Fuji ice core, The Cryosphere, 15, 5529–5555, https://doi.org/10.5194/tc-15-5529-2021, 2021.
- Passchier, C. W. and Trouw, R. A. J.: Shear Zones, pp. 111–158, Springer Berlin Heidelberg, Berlin, Heidelberg, ISBN 978-3-540-29359-0, https://doi.org/10.1007/3-540-29359-0\_5, 2005.
- Pauer, F., Kipfstuhl, S., Kuhs, W. F., and Shoji, H.: Air clathrate crystals from the GRIP deep ice core, Greenland: a number-, size- and shape-distribution study, Journal of Glaciology, 45, 22–30, https://doi.org/10.3189/S0022143000003002, 1999.
  - Raynaud, D., Jouzel, J., Barnola, J. M., Chappellaz, J., Delmas, R. J., and Lorius, C.: The Ice Record of Greenhouse Gases, Science, 259, 926–934, https://doi.org/10.1126/science.259.5097.926, 1993.
  - Rempel, A.: Englacial phase changes and intergranular flow above subglacial lakes, Annals of Glaciology, 40, 191–194, https://doi.org/10.3189/172756405781813564, 2005.
- Rempel, A. W., Wettlaufer, J. S., and Waddington, E. D.: Anomalous diffusion of multiple impurity species: Predicted implications for the ice core climate records, Journal of Geophysical Research: Solid Earth, 107, ECV 3–1–ECV 3–12, https://doi.org/10.1029/2002JB001857, 2002.
  - Ruth, U., Freitag, J., Kaufmann, P., Kipfstuhl, S., and Raynaud, D.: Continuous Measurements of Air Content: First Results from the EDML Ice Core, Antarctica, in: AGU Fall Meeting Abstracts, vol. 2004, pp. C23A–0982, 2004.
- Ruth, U., Barnola, J.-M., Beer, J., Bigler, M., Blunier, T., Castellano, E., Fischer, H., Fundel, F., Huybrechts, P., Kaufmann, P., Kipfstuhl, S., Lambrecht, A., Morganti, A., Oerter, H., Parrenin, F., Rybak, O., Severi, M., Udisti, R., Wilhelms, F., and Wolff, E.: "EDML1": a chronology for the EPICA deep ice core from Dronning Maud Land, Antarctica, over the last 150 000 years, Climate of the Past, 3, 475–484, https://doi.org/10.5194/cp-3-475-2007, 2007.
- Salamatin, A. N., Lipenkov, V. Y., and Hondoh, T.: Air-hydrate crystal growth in polar ice, Journal of Crystal Growth, 257, 412–426, https://doi.org/10.1016/S0022-0248(03)01472-6, 2003.
  - Shoji, H. and Langway, C. C.: Air hydrate inclusions in fresh ice core, Nature, 298, 548-550, https://doi.org/10.1038/298548a0, 1982.
  - Spencer, M., Alley, R., and Fitzpatrick, J.: Developing a bubble number-density paleoclimatic indicator for glacier ice, Journal of Glaciology, 52, 358–364, https://doi.org/10.3189/172756506781828638, 2006.
- Suwa, M. and Bender, M. L.: Chronology of the Vostok ice core constrained by O2/N2 ratios of occluded air, and its implication for the Vostok climate records, Quaternary Science Reviews, 27, 1093–1106, https://doi.org/10.1016/j.quascirev.2008.02.017, 2008.

- Szeliski, R.: Feature Detection and Matching, pp. 333–399, Springer International Publishing, Cham, ISBN 978-3-030-34372-9, https://doi.org/10.1007/978-3-030-34372-9 7, 2022.
- Takeya, S., Nagaya, H., Matsuyama, T., Hondoh, T., and Lipenkov, V. Y.: Lattice Constants and Thermal Expansion Coefficient of Air Clathrate Hydrate in Deep Ice Cores from Vostok, Antarctica, J. Phys. Chem. B, 104, 668–670, https://doi.org/10.1021/jp993344o, 2000.
- Tison, J.-L., de Angelis, M., Littot, G., Wolff, E., Fischer, H., Hansson, M., Bigler, M., Udisti, R., Wegner, A., Jouzel, J., Stenni, B., Johnsen, S., Masson-Delmotte, V., Landais, A., Lipenkov, V., Loulergue, L., Barnola, J.-M., Petit, J.-R., Delmonte, B., Dreyfus, G., Dahl-Jensen, D., Durand, G., Bereiter, B., Schilt, A., Spahni, R., Pol, K., Lorrain, R., Souchez, R., and Samyn, D.: Retrieving the paleoclimatic signal from the deeper part of the EPICA Dome C ice core, The Cryosphere, 9, 1633–1648, https://doi.org/10.5194/tc-9-1633-2015, 2015.
  - Uchida, T., Hondoh, T., Mae, S., Lipenkov, V., and Duval, P.: Air-hydrate crystals in deep ice-core samples from Vostok Station, Antarctica, Journal of Glaciology, 40, 79–86, https://doi.org/10.3189/S0022143000003828, 1994a.

- Uchida, T., Hondoh, T., Mae, S., Shoji, H., and Azuma, N.: Optimized Storage Condition of Deep Ice Core Samples from a View Point of Air-Hydrate Analysis, Mem. Nat. Inst. Polar Res., Special Issue, 49, 320–327, 1994b.
- Uchida, T., Shimada, W., Hondoh, T., Mae, S., and Barkov, N. I.: Refractive-index measurements of natural air–hydrate crystals in an Antarctic ice sheet, Appl. Opt., 34, 5746–5749, https://doi.org/10.1364/AO.34.005746, 1995.
- 630 Uchida, T., Hirano, T., Ebinuma, T., Narita, H., Gohara, K., Mae, S., and Matsumoto, R.: Raman spectroscopic determination of hydration number of methane hydrates, AIChE Journal, 45, 2641–2645, https://doi.org/10.1002/aic.690451220, 1999.
  - Uchida, T., Miyamoto, A., Shin'yama, A., and Hondoh, T.: Crystal growth of air hydrates over 720 ka in Dome Fuji (Antarctica) ice cores: microscopic observations of morphological changes below 2000 m depth, Journal of Glaciology, 57, 1017–1026, https://doi.org/10.3189/002214311798843296, 2011.
- Ueltzhöffer, K. J., Bendel, V., Freitag, J., Kipfstuhl, S., Wagenbach, D., Faria, S. H., and Garbe, C. S.: Distribution of air bubbles in the EDML and EDC (Antarctica) ice cores, using a new method of automatic image analysis, Journal of Glaciology, 56, 339–348, https://doi.org/10.3189/002214310791968511, 2010.
  - van der Walt, S., Schönberger, J. L., Nunez-Iglesias, J., Boulogne, F., Warner, J. D., Yager, N., Gouillart, E., Yu, T., and the scikit-image contributors: scikit-image: image processing in Python, PeerJ, 2, e453, https://doi.org/10.7717/peerj.453, 2014.
- 640 Weikusat, I., Kipfstuhl, S., Faria, S. H., Azuma, N., and Miyamoto, A.: Mean grain size of ice core EDML, PANGAEA, https://doi.org/10.1594/PANGAEA.807132, in supplement to: Weikusat, I et al. (2009): Subgrain boundaries and related microstructural features in EDML (Antarctica) deep ice core. Journal of Glaciology, 55(191), 461-472, https://doi.org/10.3189/002214309788816614, 2009.
- Weikusat, I., Lambrecht, A., and Kipfstuhl, S.: Eigenvalues of crystal orientation tensors for c-axes distributions of vertical thin sections from the EDML ice core, https://doi.org/10.1594/PANGAEA.807142, 2013.
  - Weikusat, I., Jansen, D., Binder, T., Eichler, J., Faria, S. H., Wilhelms, F., Kipfstuhl, S., Sheldon, S., Miller, H., Dahl-Jensen, D., and Kleiner, T.: Physical analysis of an Antarctic ice core—towards an integration of micro- and macrodynamics of polar ice, Philosophical Transactions of the Royal Society A: Mathematical, Physical and Engineering Sciences, 375, 20150 347, https://doi.org/10.1098/rsta.2015.0347, 2017.
- Wilhelms, F., Miller, H., Gerasimoff, M. D., Drücker, C., Frenzel, A., Fritzsche, D., Grobe, H., Hansen, S. B., Hilmarsson, S. A., Hoffmann, G., and et al.: The EPICA Dronning Maud Land deep drilling operation, Annals of Glaciology, 55, 355–366, https://doi.org/10.3189/2014AoG68A189, 2014.