# Peer review of "Air clathrate hydrates in the EDML ice core, Antarctica"

_EGUsphere, 2025_

## Author Response (AR1)

**EGUSPHERE-2025-633**

**Air clathrate hydrates in the EDML ice core, Antarctica**

Referee report #1

**General comments**

The manuscript presents the first experimental data on the air-hydrate geometrical properties in the EDML ice core, below the bubbles-to-hydrates transition, which were obtained by automated image analysis of microphotographs of thick sections of ice taken in the field a few days after core retrieval. The resulting data set represents a significant addition to the existing experimental data on the geometric properties of air hydrates in polar ice sheets, which until recently were obtained only from GRIP, Vostok, and Dome Fuji ice cores. Specific conditions of the EDML site on the Antarctic Plateau (relatively warm temperature of ice and enhanced accumulation, location on the ice divide) allowed authors to reveal some new peculiarities of hydrate properties such as weakening of their correlation with the climate (isotope content of ice) on small time scales, and a distinctive change in air hydrate properties associated with changes in the deformation regime of ice reflected in its microstructure. The most important achievement of this work is a carefully developed technique of using automatic analysis of high-quality images of thick sections of ice in order to obtain various geometric characteristics of air-hydrate inclusions in polar ice. This technique was, for the first time, successfully applied to quantitatively describe air hydrates in an Antarctic ice core. Its further refinement will provide a reliable tool for the study of future ice cores, including those that will be obtained in Antarctica as part of the Oldest Ice project. To summarize, I believe that this work deserves to be published in TC after the small corrections and clarifications suggested below are made by the authors.

We want to thank the reviewer for the thorough review and especially the additional insights that will improve the manuscript. We reply to the specific comments and technical corrections below.

Concerning the presentation of the material, I would like to encourage the authors to make an additional effort to improve the structure of the text in order to make it easier to read and understand. To me, the most obvious idea in this regard would be to move sections 4.1 and 4.2 of the ms from Discussion to Results, and to merge them with sections 3.1 and 3.2, respectively. Some of my specific comments below also reflect the difficulties I encountered in reading the manuscript and therefore, hopefully, may help to improve its presentation.

Our aim was to clearly separate measured results and the discussion of the measured values. We understand that this might make it more difficult to read and could lead to a misinterpretation of the message of the manuscript.

We will improve the structure of the text and especially try to better differentiate between method-related results and measured results. For this, we will follow the reviewer's advice and move sections 4.1 and 4.2 from Discussion to Results, and merge them with sections 3.1 and 3.2, respectively. I.e. we move section 4.1 to 3.1.1 and the first paragraph of 4.2. to 3.2.1.

However, L263 – L273 should not be merged with section 3.2. We propose to merge these lines with section 4.3.1 and add them after L301.

This should hopefully improve the clarity and presentation of the manuscript.

**Specific comments**

L15-17: The BHTZ depth range depends on the temperature in the ice sheet (e.g., 500-1250 m at the cold Vostok site, but 1000-1500 m at the GRIP drilling site, which is warmer). What site does the 500-1500 m depth interval refer to? I did not find in the ms clear information about the BHTZ depths at the EDML site, which is important for interpreting the bubble and hydrate data.

The 500 – 1500m depth interval mentioned includes the minimum depth of the BHTZ (Vostok) and the maximum depth (GRIP) observed for deep ice cores. We agree that this could lead to misunderstandings. We will delete the depth range from L15-L17 and clearly state the extend of the BHTZ for EDML (700m to 1225m) at L41 where we introduce the EDML ice core.

At L249 one reads: "Note that LGM-ice at EDML coincides with the top of the BHTZ", which is a bit misleading, since the LGM ice at EDML is buried at a depth of about 1000 m. According to Bendel et al. (2013), the BHTZ at EDML is between 700 and 1200 m and therefore LGM ice coincides with the middle of the transition zone, which disrupts the correlation of bubbles' properties with climate. Please state clearly the BHTZ depths at EDML in the introduction. It would also be useful to give in the **introduction** the maximal depth (from the ice core log) of the borehole when it reached the bedrock (at 2774.15 m?).

We agree that our wording is not very precise. We will change:

"Note that LGM-ice at EDML coincides with the top of the BHTZ, therefore, air bubbles are already starting to convert to air hydrates."

to

"Note that LGM-ice at EDML is located within the BHTZ (at about 1000 m depth), therefore, air bubbles are already converting to air hydrates."

In addition, we will add information about the BHTZ at EDML in the Introduction as well as the maximum logging depth. It is indeed 2774.15m (Wilhelms et al., 2014; https://doi.org/10.3189/2014AoG68A189)

L28-30: To the best of my knowledge, the most recent review on this topic is given in Lipenkov V. Y. How air bubbles form in polar ice, Earth's Cryosphere, 2018, 22, 16–28.

We thank the reviewer for the introduction to this very useful publication. We will add it to our references.

L186-188: For hydrate inclusions with non-isometric shape and preferred orientation in space, it is also important to choose the right plane for the thick section analysis (i.e. image

plane) that would allow proper assessment of inclusions' dimensions and aspect ratios. Have you tried to solve this problem, and if so, how?

If the objects have a preferred orientation, then the "perfect-observation" plane would be the one in which the objects show the maximum elongation. With our technique we can unfortunately not make sure we are measuring the maximum elongation.

We tried to estimate the change in projection properties of an ellipsoid that is rotated around to core axis, but decided not to include it in the manuscript.

Below are the results of three end-member cases. On the left side, the ellipsoid together with the projection to the XZ-plane, which corresponds to the plane of the microphotograph, can be seen (c.f. Fig.1 in the manuscript). The plots on the right side show the evolution of the projection properties with progressing rotation around the Z-axis (core axis). a (red), b (green), c (blue) are the ellipsoid axis and the orientation is the angle (-17°) between the semi-major axis (a) and the X-axis. 0° on the x-axis of the plots represents the initial condition as seen on the left image.

In summary, the challenge lies in the fact that we do not know if we measured the maximum elongation and that the third dimension is unknown, i.e. if the non-isometric object is more cigar (prolate)-like or more oblate-like.

For our interpretation of the air hydrate shape characteristics, we consider the relative orientation of the samples to each other. As mentioned in section 2.5.2., a change in relative orientation could be caused due to a difficulty in matching core breakpoints during ice-core logging.

Weikusat et al., (2017) report a loss in azimuthal orientation between 1686m and 1696m as well as between 1955m and 2035m depth.

Between 1665m and 1695m, we observe an increase in air hydrate median inclination from 17° to 22° as well as a "polarity" change. Below are the orientation plots of the air hydrate major axis ("rose plots"), "n" corresponds to the number of measured objects.

However, we do not observe a significant change in the median AR from 1665m to 1695m depth.

Between 2005 and 2035m depth, we observe a rather continuous decrease in the median inclination (16°, 14°, 10°) and increase in AR (1.38, 1.4, 1.46). Below are the orientation plots displaying the continuous decrease in orientation.

The fact that the median AR does not change between 1665m and 1695m, while the orientation does (Fig. 7b), implies that either the air hydrates are not very well sorted, or that the majority is rather isometric in shape.

On the contrary, the continuous and joint change of AR together with orientation from 2005 to 2035m depth (Fig. 7b) point either towards an increase in sorting or a decrease in isometric air hydrates. Both could be induced by the increase in simple shear deformation.

In the future, this problem could be addressed with taking images from two (or more) different directions.

If the reviewers and the editor deem it appropriate, we would suggest including this information in the Appendix.

L196 "Surprisingly, no air hydrates were found in the lowermost sample at about 2774 m depth": Any comment on this: distance of this sample from the bedrock, measured air content in it, accreted ice?

This question is related to a question asked by reviewer#2 (196). We will provide identical answers to the two related questions:

The lowermost sample is from 2773.91 m depth. This means that it is about 25 cm above the bottom of the ice core. To the best of our knowledge there are no air content measurements for this sample. However, it was reported that, from 2760 m, the concentration of micro inclusions gradually decreases with depth (Faria et al., 2010). The bottom of the core reaches the pressure melting point (-2°C; Faria et al., 2010) and ice core drilling was terminated because subglacial water was entering the borehole (Wilhelms et al., 2014).

Therefore, it could either be accreted ice with no, or very little impurity inclusions (at least in the 10cm sample!). Or meteoric ice that underwent processes which resulted in the expulsion of impurities and gases, for instance the segregation of impurities to the grain boundary network and the subsequent drainage to the bedrock driven by the hydraulic gradient (Rempel 2002,2005). These processes are considered to alter the basal ice at EDC, Dome Fuji and NEEM (Tison et al., 2015; Ohno et al., 2016; Goossens et al., 2016).

While we currently cannot explain its origin, we think that the information about the air hydrate content in the deepest sample is an interesting detail that deserves mentioning. Further investigations of the deepest ice at EDML are necessary to determine its origin and explain the absence of air hydrates.

We will add this discussion about the last sample to the end of paragraph 4.3.2

Rempel et al., 2002: https://doi.org/10.1029/2002JB001857

Rempel 2005: https://doi.org/10.3189/172756405781813564

Faria et al., 2010: https://doi.org/10.1016/j.quascirev.2009.10.016

Tison et al., 2015: https://doi.org/10.5194/tc-9-1633-2015

Goossens et al., 2016: https://doi.org/10.5194/tc-10-553-2016

Ohno et al., 2016: https://doi.org/10.1002/2015JF003777

L68-69 "For this study, mosaic images located about 1.5 mm below the sample surface were analyzed..."; L179 "We convert the measured air hydrate counts per sample to Nah using the measured ice-sample thickness..."; L180-182 "Air hydrate counts per sample and Nah have a good linear correlation (Fig. 4) and we conclude that the observed volume (i.e. hydrate mapping depth; Fig. 1) is consistent for all samples"; L253-254 "... the hydrate-mapping depth in our microscopic set-up might not correspond to the physical sample thickness (Fig. 1). In other words, the microphotographs from one focus plane might not display the entire sample volume". The construction of the narrative is such that it is not until L253-255 that the reader begins to realize that you could not have obtained Nah (in cm-3) because you did not know the thickness of the ice layer (the hydrate mapping depth) within which the hydrate counts were made. Why not say this at the very beginning, and treat the hydrate count data as a reliable but relative metric of Nah (assuming the observed volume was the same in all samples), and then scale that metric with the measured air content of ice, and use the thus derived Nah in further consideration? (Such a relative metric of Nah can be the density of hydrates' projections (cm-2) observed over an 80 x 30 mm area of the mosaic image of the sample).

As stated above, our aim was to clearly separate the description of measured results and the discussion of the measured values. The conclusion, that we did not know the thickness of the hydrate mapping, depth is derived from discussing our measured values with available literature. We agree with the reviewer that this could lead to a misinterpretation of the data.

To address this, we will follow the reviewer's suggestion regarding the presentation of the material as explained on page one.

Regarding the conversion of air hydrate counts to Nah, we believe that using the measured sample thickness represents the most accurate values by making the least assumptions. I.e. we stay closes to the real appearance.

Figure 5 caption: "Error-band for b) is explained in section 2.4". I could not find an estimate of the resulting Nah error in Section 2.4. In any case, I now suspect that this resulting error shown in Fig. 5 does not account for the difference between the sample thickness that was used to calculate Nah and the hydrate mapping depth.

We agree. The orange error band in Figure 5 is displayed for the air hydrate counts and is derived from the analysis of the "ground truth" (manually segmented) images. The values used are given in Table 2.

We will add error bars to the Nah values in Figure 5 including the uncertainty in thickness measurements.

L226 "Naturally, the TACh and Vc signals show the same pattern (Fig. 6b)". Not sure that this statement matters when both TACh and Vc are shown by a single graph. Delete the sentence?

We agree that this sentence may be redundant, however, we added it to emphasize the direct connection of air hydrate Vc and TACh, which might not be immediately obvious to an audience not very familiar with this topic. This was confirmed by a question of referee #2, we thus decided to keep it in the manuscript.

L237-241: It would be appropriate here (or elsewhere) to provide information on the inclination of the borehole from which the studied ice core was obtained.

The inclination of the borehole was measured in 2005 and is roughly between  $+/-3^{\circ}$  (Weikusat et al., 2017; https://doi.org/10.1098/rsta.2015.0347). We will add this information to the manuscript in the Introduction at L38.

L247-248: In Fig. 3 from Bendel et al. (2013), the average Nab value in LGM ice appears to be closer to 400 cm-3.

We agree. 500 cm-3 represents the lowest measured value, but most values are closer to 400 cm-3. We will consider this together with the comment below (L250-251).

L250-251: Number concentration of air bubbles and that of succeeding hydrates (if we assume one-to-one conversion) depends on the ice formation conditions (accumulation rate, firn temperature and surface snow density), so the properties of the Vostok ice cannot be simply projected onto the EDML ice core. Using a simple model, as described in Lipenkov (2018), one can estimate Nab ~400 cm-3 for present-day conditions at EDML (-44.5 °C; 6.4 g/cm2 yr; 0.38 g/cm3) and Nab ~475 cm-3 in the LGM ice (assuming: -54 °C; 3 g/cm2 yr; 0.38 g/cm3). The model estimate for Holocene ice is close to Bendel's et al. data while that for the LGM ice is slightly higher than their experimental data (because LGM ice coincides with the transition zone), and lower than the Vostok-based value used in the ms (the extremely low LGM temperature at Vostok led to a very small grain size at close-off and hence a great number of bubbles forming).

We agree with the reviewer that the Nab properties of the Vostok ice core cannot be fully projected to the EDML ice core. Our estimation pairing the observations at EDML (300-400 cm-3 Nab) with observations at Vostok (factor of 1.7) would have represented a "worst-case" with respect to Nab for LGM conditions.

We want to thank the reviewer for the very useful information regarding the Nab estimations using the model described in Lipenkov (2018). Especially for the LGM conditions, where "true" Nab of EDML cannot be measured, they provide a better estimation compared to the "worst-case" we used previously. We will implement the new estimations of

- Nab ~400 cm-3 for present-day conditions at EDML (-44.5 °C; 6.4 g/cm2 yr; 0.38 g/cm3)
- Nab ~475 cm-3 in the LGM ice (assuming: -54 °C; 3 g/cm2 yr; 0.38 g/cm3)

in our manuscript. We will then recalculate the detection rate of air hydrates based on estimations from air bubble content.

L259: It appears that the CFA system used by Ruth et al. (2004) does not provide accurate absolute values of air content (it was designed primarily to document high-resolution relative TAC variations). For that reason, the obtained average value (0.0815 cm3/g) is lower than the TAC measured with absolute methods in the Holocene ice cores from the most elevated drilling sites in Antarctica (Dome F, Vostok, EDC). For the EDML elevation (2892 masl) and atmospheric pressure (~700 mb) we can expect the TAC to be slightly above 0.09 cm3/g (Martinerie et al., 1992) in Holocene ice and even higher in LGM ice. This prediction has been confirmed by the measurements made with an absolute barometric method in LGGE/IGE, Grenoble, which gave TAC=0.0906±0.0025 cm3/g in Holocene ice, 0.0970±0.0020 cm3/g in LGM ice, and a mean value equal to 0.0924±0.0031 cm3/g (unpublished data). I suggest that the authors use estimates based on Martinerie, P., Raynaud, D., Etheridge, D.V., Barnola, J.-M., Mazaudier, D. Physical and climatic parameters which influence the air content in polar ice. Earth Planet. Sci. Lett. 1992, 112: 1-113, rather than the experimental data from Ruth et al. (2004).

The values from Ruth et al. (2004) were, to the best of our knowledge, the most direct TAC measurements for EDML. We highly appreciate the reviewer's insights on the unpublished measured TAC for the EDML ice core which agree with the estimations by Martinerie et al., 1992. Or rather, the theory agrees well with the measurements. We will use the estimates for TAC based on Martinerie et al., 1992, as suggested by the reviewer, to replace the obtained average value from Ruth et al. (2004).

L260: It may be noted here that uncertainty in hydrate size determination can also be the cause of the observed mismatch between calculated and measured air content (Vc is very sensitive to even small errors in hydrate linear dimensions), but in this particular case the greatest contribution to this mismatch comes, of course, from the uncertainty in Nah.

We agree that the Vc is very sensitive to even small errors in determining the air hydrate size. We will add this information to the manuscript.

We further discuss this point for the reviewer's comments for L269-273, L310-312 and L312-313.

L267: Replace "air hydrates dissociating" with "air hydrates dissolving".

We will replace "air hydrates dissociating" with "air hydrates dissolving".

L269-273: This reasoning seems a bit odd to me, because: 1) below the BHTZ, almost all of the air trapped by ice (98% or so) is stored in air hydrates. Therefore, whatever evolution the hydrate system undergoes (growth of larger hydrates at the expense of smaller ones, coalescence, etc.), the geometric properties of the hydrate ensemble, if accurately measured, should allow us to correctly estimate the TACh; 2) if the cage occupancy in Eq. (9) is

erroneously overestimated, the calculated TACh must also be overestimated, but it seems to show the opposite trend?

1) In principal we do agree with the reviewer that almost all of the air trapped in ice is stored in air hydrates. However, as the reviewer states in L260, Vc is very sensitive to even small errors in hydrate linear dimensions. We do have uncertainties in the measurements, which increase with complex-shaped and clustered AH in the images. In the depth region just below the BHTZ we do observe more complex-shaped AH compared to deeper parts, which partly seem to show recrystallization towards an equilibrium state (regular and rounded morphologies). Below, some examples for complex-shaped AH can be seen which seem to be in a transition stage to a more regular object.

|                 | 0               |                 |                 | 9               |                 |
|-----------------|-----------------|-----------------|-----------------|-----------------|-----------------|
| 1285m           | 1285m           | 1355m           | 1355m           | 1405m           | 1566m           |
| $D = 286 \mu m$ | $D = 189 \mu m$ | $D = 325 \mu m$ | $D = 247 \mu m$ | $D = 270 \mu m$ | $D = 202 \mu m$ |
| Cloudy with     | Disc-like,      | Cloud-like;     | Disc-like,      | Hollow ring     | Cloudy with     |
| protrusion      | flat ?          | hollow?         | flat ?          |                 | protrusion      |

We refer here to Pauer et al. (2000), Kipfstuhl et al. (2001) and Weikusat et al. (2015), who discuss air hydrate metamorphosis / recrystallization and their different morphologies.

Pauer et al., 2000: doi:10.2312/polarforschung.68.267

Kipfstuhl et al., 2001: https://doi.org/10.1029/1999GL006094

Weikusat et al., 2015: https://doi.org/10.3189/2015JoG15J009

2) From 1255 – 1700m the TACh decreases from 0.085 cm3g-1 to an average of 0.059 cm3g-1. This means the TACh is overestimated relative to the average value. A lower cage occupancy would lower the TACh. The variations in cage occupancy with different formation conditions however, are not large enough to solely explain the overestimated TACh.

E.g. Uchida et al., 1999: https://doi.org/10.1002/aic.690451220

Hachikubo et al., 2022: https://doi.org/10.1021/acs.jced.2c00602

We will modify the manuscript in L263-273 as follows and include the table above as a new figure:

"From 1255 m to 1700 m (spanning about 26 kyr), the T ACh decreases from 0.085 cm3 g-1 to 0.059 cm3 g-1. For the same depth region, we observe a decrease in the RSD of the air hydrate D distribution (Fig. 5c) and a strong decrease in the AR 90% percentile from 2.5 to 2 (Fig. 7c). One explanation for the overestimation of the T ACh (relative to the average value

of 0.059 cm3 g-1) could be the fact that the air hydrates' true cage occupancy for this region is probably slightly lower than our estimated 0.9, as it depends on pressure and temperature (Chazallon and Kuhs, 2002). However, the variations in gas hydrate cage occupancies with different formation conditions (e.g. Uchida et al., 1999; Hachikubo et al., 2022) are not large enough to solely explain the observed deviation from the mean.

Another reason could be an uncertainty of the air hydrate size, as the calculation of Vc is very sensitive to errors in size determination. In the depth region just below the BHTZ we observe more complex-shaped air hydrates (Fig. ) compared to deeper parts, which is indicated in the data by the decrease in the RSD and the AR 90% percentile (Fig. 5, 7c). The complex-shaped air hydrates partly seem to show a recrystallization towards an equilibrium state (regular and rounded morphologies).

For the Vostok ice core, Lipenkov (2000) reported a depth region of faster air hydrate crystal growth rates, extending about 300 m below the BHTZ (i.e. down to 1550 m), and explains this by the large number of small, oxygen-enriched air hydrates dissolving in this region. Down to the same depth, Suwa and Bender (2008) measured noisy  $\delta$ O2/N2 ratios. Similarly, Oyabu et al. (2021) reported a large scatter in measured  $\delta$ O2/N2 ratios for about 300 m (spanning 25 kyr) below the BHTZ for the Dome Fuji ice core. Therefore, we surmise that the region from about 1255 - 1700 m at EDML coincides with a region of relatively faster air hydrate growth and recrystallization and the air hydrate ensemble might not have reached an equilibrium state. In summary, the overestimation likely originates from uncertainties of measurements in air hydrate size, which increase with complex-shaped air hydrates in the images."

L286-289: The geometric properties of air hydrates (inherited from the properties of bubbles) depend on both the temperature and accumulation rate prevailing during ice formation. The correlation between accumulation and temperature (isotope content of ice) on small time scales is not as obvious as in the case of global glacial-interglacial climate changes, and may not exist at all. This may also be the reason for the weakened correlation between Nah and  $\delta 180$  in the case of smaller-scale climatic fluctuations.

We want to thank the reviewer you for this comment. We will add this information to L287 where we discuss possible reasons for the poorer correlation between Nah and d18O on smaller timescale compared to glacial-interglacial transitions.

L299-300: I would rewrite as "Due to the greater surface curvature, smaller particles are more soluble than larger particles..."

We agree that this wording provides more clarity and we will we adopt it into our manuscript.

L310-312: With the exception of two data points, I do not see particularly large mean hydrate diameters in region 2 in Fig. 5c. It would be good to say here how the inability of your image analysis to distinguish individual air-hydrate crystals within their clusters may affect the count of hydrate number? It seems like it should lead to an underestimation of the number of individual hydrates (although the hydrate number concentration shows anomalously high values in zone 2).

We agree with the reviewer that the mean diameters in region 2 are not particularly large. However, the mean diameter is slightly overestimated considering that the samples from

region 2 correspond to a cold period (probably MIS 6) and considering the linear relation of Nah and mean diameter (Fig. 8a). We also agree that the inability to distinguish individual air hydrate crystals within clusters leads to an underestimation of the air hydrate number. The quantification of the underestimation, however, is difficult.

We will add the following paragraph to the manuscript at L313:

"Besides the overestimation in volume, the increase in air hydrate clusters also leads to an underestimation of Nah. However, quantification is challenging due to the difficulty in distinguishing individual air hydrate crystals within clusters from the images."

L312-313: I cannot agree with this explanation for the abnormal volume concentration and TACh. Whatever the mechanism of hydrate coalescence, this process cannot change the initial (pre-coalescence) volume concentration of hydrates (and TACh), since this property of ice is as constant as its air content. As for the quantitative assessment of the geometrical properties of all hydrate inclusions (without discrimination into clusters and individual hydrate crystals), one would expect an underestimation of the size of hydrate clusters rather than an incorrect count of the total number of hydrate inclusions (individual crystals and their clusters), which should lead to an underestimation of the calculated total volume concentration of hydrates, but not the other way around. The underestimation of cluster size in image analysis could be quite expected due to the partial overlap of the projections of two or more individual hydrate crystals included in the cluster (see Fig. 7c) and the difficulty in assessing the cluster size in the direction normal to the image plane.

We agree with the reviewer that the volume concentration (and TAC) are very unlikely to be increased by a physical process, i.e. it is unlikely that a concentration of air hydrates, and therefore a concentration of TAC, is happening. We realize that we have to improve our explanation in the manuscript.

The abnormally high values simply result from our way of estimating the volume of the air hydrates. The diameter of an air hydrate is defined as the diameter of a circle having the same area as our measured object. For instance, if we measure a cluster that has 4 times the area of a non-cluster, the diameter would be doubled (e.g. Fig 9). The volume however, would be 8 times the one of a non-cluster.

The total volume of 4 smaller air hydrates (V1) compared with the total volume of 1 larger air hydrate cluster (having the same area as the 4 smaller ones combined; V2), leads to an overestimation of V2. In this particular case V2 = V1 \* 2. The relationship of V1 and V2 depends on the amount of air hydrates per cluster.

In practice, the true overestimation is difficult to assess due to the uncertainty in the amount of air hydrates per cluster and the degree of overlap of the air hydrates.

We will modify the sentence in L312: "Furthermore, the emerging of the latter explains the high mean volume and the abnormal volume concentration and TACh, as the volume of the cluster is overestimated during calculation of the volume from the area assuming a spherical shape (Fig. 6a,b)."

L326-327: It is not quite clear what phenomenon the authors have in mind, but if it is the disappearance of the climate-related variations in the hydrate properties in the course of Ostwald ripening, then the fundamental reason for this is that the hydrate growth rates are inversely proportional to the square of the mean initial hydrate size, which leads to dumping of the climatically induced variations in the hydrates' geometrical properties (Salamatin et al., 2003).

We agree with the reviewer that the fundamental reason for the dampening of the climate related fluctuations of Nah and mean size is that the growth rates of air hydrate ensembles with smaller mean size are higher than growth rates for ensembles with a larger mean size (Salamatin et al., 2003).

Increasing age of the ice, temperature and pressure are parameters connected to the fundamental reason.

We will add this information in L326 and change "The fundamental parameters behind this phenomenon are the increasing depth (and pressure), increasing ice temperature, and increasing age of the ice (Uchida et al., 2011)" to to "The fundamental reason behind the disappearance of climate-related variations in Nah and D is the fact that air hydrate growth rates are inversely proportional to the cube of their mean initial size (Salamatin et al., 2003).

In other words, the growth rates of air hydrate ensembles with smaller mean size are higher than growth rates for ensembles with a larger mean size. Other parameters that influence air hydrate growth are increasing depth (and pressure), increasing age and increasing ice temperature (Uchida et al., 2011)."

**Technical corrections**

The caption to Fig 5 needs some explanation of the zones 1 and 2 shown in the figure.

We will add the explanations for region 1 and 2 to Fig. 5 and Fig. 7.

Is the subscript 0 at n0 really needed in Eq. (9)?

The Loschmidt constant is usually given as n0 or sometimes as Nl. We will change n0 to Nl to better distinguish it from the number of cages per air hydrate unit cell (n).

L207: Reference to Fig. 8 is given before references to Figs. 6 и 7. Consider changing the order of the figures.

We will consider changing the order along with restructuring certain paragraphs as suggested by the reviewer.

In Fig. 6, the blue and green symbols are not well distinguishable on my screen. Consider changing the colors.

We will change the colors to blue and orange to make it better distinguishable.

Best regards,

Florian Painer et al.

**EGUSPHERE-2025-633**

**Air clathrate hydrates in the EDML ice core, Antarctica**

**Referee report #2**

The authors present a new method for analyzing clathrate hydrates using a mosaic-style automated image processing method in "thick sections" of ice-core ice...specifically in their study using the EDML core. They note that this technique allows for more automation and analysis of multiple clathrates while reducing (often tedious and time-consuming) manual counting. The authors used images that were taken almost immediately after core recovery, thereby reducing the risk of clathrate decomposition as a factor in their measurements/analyses.

We thank the reviewer for the encouraging evaluation of our manuscript. The comments and questions will help us to improve the clarity and quality of the manuscript. We reply to the specific comments below.

One of the biggest questions I had was regarding the interpretation of calthrate numberdensity from a 2D image. The authors do note that they used the recorded thicknesses of the "thick sections" to calculate appropriate volumes for each sample, but that they also used mosaic images from a specific 2D plane in the samples (1.5 mm). This would seem to present some stereological considerations. For example, once clathrate counts are completed for a sample, when determining the appropriate volume for that sample, are cut corrections factored in? Should it be assumed that, similar with ice core bubbles in "thick sections", when sections are planed (or microtomed) for imaging, that some of the clathrates are cut at the surfaces? Given this, an additional amount of "volume" needs to be added back in to account for the portion of the clathrate (or bubble) that was removed. Martinerie and others (1999) as well as Saltykov (1976) note this and base the correction on both the shape of the sample as well as the mean size of the feature being measured in that sample. The end result usually means slightly lower number-densities than first measured. This interpretation of thick sections also leads to considerations when looking at shapes or other orientation data on features as it's not always clear which axis the observer is looking down. Fegyveresi and others (2019) note this to some extent, and made some notable assumptions.

Air hydrates cut at the surface of the ice sample decompose within hours. They leave behind a void that can be seen in the images as black objects (similar to air bubbles). We take care of this by filtering objects with a low mean gray value (i.e. dark objects). Therefore, we conclude that we do not need to consider an additional amount of volume to correct for cut air hydrates.

Fegyveresi and others (2019) correctly note that the maximum elongation is only represented for bubbles elongated in the plane of section. This is also true for air hydrate measurements. Similar to Fegyveresi and others (2019), our technique did not allow us to measure, or identify, the full air hydrate elongation.

Fegyveresi and others (2019) partially overcome this problem by measuring bubble elongation in grains with c-axes close to the plane of section. Provided that air hydrates

preferentially elongate along the basal plane of ice, which has yet to be confirmed, this could be an approach to estimate maximum air hydrate elongations in the future.

Because the number of clathrates is ostensibly determined by the number of bubbles, which in turn are determined by the number of grains, and ultimately the site temperature and accumulation rate, it may be beneficial to show a comparison of Nah to bubble number-density with depth. Visually it would also be interesting to see the crossover of bubble decrease and clathrate increase in the transition zone. It would also give some preliminary estimates of bubble-to-clathrate ratios.

We agree that it would be great to show the evolution of Nab before- and in the BHTZ and compare it the with increase in Nah in the transition zone. However, or manuscript focusses on air hydrates below the BHTZ. Assessing of air hydrate properties in the BHTZ could be addressed in future work.

In the calculations/estimations of Total Air content, (ln ~260-275), are there any possible surface elevation implications that could be interpreted from changes in TAC, or perhaps changes in impurity content?

The TACh (Total air content included in air hydrates) obtained with our method should be primarily regarded as a control parameter for the accuracy of the method. We thereby use a similar approach to Lipenkov (2000). Our estimates are currently too uncertain to draw physical conclusions on surface elevation, etc.

We understand that our decision to place paragraph 4.2. in the Discussion can lead to a misinterpretation. We will follow the suggestions by reviewer #1 and move sections 4.1 and 4.2 from Discussion to Results, and merge them with sections 3.1 and 3.2, respectively. I.e. we move section 4.1 to 3.1.1 and the first paragraph of 4.2. to 3.2.1. However, L263 – L273 should not be merged with section 3.2. We propose to merge these lines with section 4.3.1 and add them after L301.

**This should improve the clarity of the manuscript.**

Are there any other ice-flow or deformational considerations that need to be made with respect to the site and observed measurements? Has the ice divide migrated over time? has the ice flow direction changed? What about borehole inclination or azimuth uncertainty of thick sections with respect to ice flow? Was ice-flow azimuth of the cores even recorded during recovery?

It is currently unclear whether and how the ice divide(s) migrated and if the ice flow direction changed through time. Generally, the ice at EDML most likely originates from further upstream around the Dome Fuji region (e.g. Huybrechts et al., 2007; https://doi.org/10.5194/cp-3-577-2007). This can cause non-climatic biases, e.g. in the interpretation of the d18O record (temperature). For this work, we decided to use measured d18O data rather than reconstructed values, especially since reconstructed values are not available for samples deeper than 2415m.

The borehole inclination was measured in the year 2005 and is roughly between  $\pm$  (Weikusat et al., 2017; https://doi.org/10.1098/rsta.2015.0347). We will add this information to the manuscript in the Introduction.

Unfortunately, the absolute orientation of the samples with respect to ice flow could not have been obtained.

We thank the reviewer for the additional interesting questions. However, further interpretation of our data are beyond the scope of this manuscript.

Figure 5, panel 1, are the d18O data different colors because they come from two sources? (Also, the caption the Meyer et al. has no date.) While they are referenced in the text, the shaded bands should also be noted in caption. It's also not entirely clear what "a pronounced deviation of climatic influence" means here. Is this specific to the d18O data?

1) The blue data correspond to published record (EPICA Community Members, 2010) for the dated part of the EDML ice core (down to 2415m depth). Below 2415m depth, due to large scale disturbances in the ice stratigraphy, the integrity of the d18O record is lost. The different coloring should emphasize this.

They gray data were first mentioned by Weikusat et al. (2017) (personal communication with H. Meyer and H. Oerter 2016). They will be made publicly available via the open-access repositories PANGAEA by Meyer et al..

We will modify the figure caption for Fig. 5,7 as follows: "a) Paleo-climatic information as  $\delta$ 18Oice values. Blue corresponds to the dated- (EPICA Community Members, 2010), gray to the non-dated part of the ice core (Meyer et al.),

2) Yes. We use the d18O data as a proxy for past climate to compare it with measured air hydrate number and size. We will change L291 – 294 to make this clearer. "For samples from 2025 - 2115 m depth (region 1 in Fig. 5), encompassing MIS 5b and parts of MIS 5c, we observe an evident deviation of the climatic influence on the air hydrate ensemble. We measure a low Nah and a relatively large mean diameter (c.f. section 4.4.1), whereas the colder climate during MIS 5b (Fig. 5a) should result in a relatively high Nah and a small average size of air hydrates."

Figure 6 - I had some issues with the two colors being so similar...maybe just swap for something different? Perhaps noting in the graphs that the green dots are "Scaled values" instead of "adjusted". ...or maybe Nah-derived values (w 1.4 scaling applied). Either way, it should be clearer which color goes with which axis.

Thank you for this suggestion, we will change the coloring for this figure and replace "adjusted" with "scaled".

In this case, both colors correspond to both axis as the volume concentration can be directly converted to the TACh. We wanted to make this clear with the sentence on L226: "Naturally, the TACh and Vc signals show the same pattern (Fig. 6b)."

Figure 7 the eigenvalue lambda label should also be labeled on the axis or identified in the caption for the reader. Also, what are the shaded bands 1 and 2?

We will add the lambda label to the caption and add the explanation for the shaded bands Fig. 5 and Fig. 7.

Figure 7 also shows a single max c-axis at ~2230, yet the text says it happens at 2030m.

For this Figure, the change in ice CPO via depth at EDML is indicated with the change in the second eigenvalue (lambda 2) of the orientation tensor of the ice crystals c-axis distribution. Lambda 2 shows the development of a single maximum at 2030m, which holds down to about 2500m. The insets of the schematic stereographic projections are only meant to visualize the change in CPO corresponding to the three "terraces" visible for the lambda 2 parameter. We think that using all 3 lambda values would overload the Figure, but the complete CPO evolution at EDML is given by Weikusat et al. (2017).

Weikusat et al., 2017: https://doi.org/10.1098/rsta.2015.0347

We will make this clearer in the Figure caption.

Figure 8 - can you re-identify the stars? I know they are noted in previous plots, but it would be helpful to identify here as they are so noteworthy.

We agree and will reidentify the stars.

250 Is there a physical significance to the 1.4 scaling, or was it simply the number that allowed for the Nah values to correlated with measured TAC? Also why was this not shown in the graphs?

There is no direct physical meaning for the scaling factor. It was used to match the mean TACh with the mean TAC measured by Ruth et al. (2004) and to estimate the measurement accuracy. We decided not to include the scaling factor in the other graphs because we want to stay close to the actual measured values.

The improvement of the structure of the text should provide additional clarity (see answer to L260-275).

330 Authors note: For the EDML ice core, the age of the ice deeper than 2415 m is not known. However, the sudden increase in ice grain size by one to two magnitudes to up to 0.5 - 1 m for the deepest samples (Faria et al., 2010a; Weikusat et al., 2017) and the relatively warm in-situ ice temperature of at about 2550 m (Figure 7d) indicate favorable conditions for air hydrate crystal growth. Thus we interpret our data to mean that from this depth, the climate signal is dampened by considerable air hydrate crystal growth." - So should it be assumed then that there is a trend that would need to be accounted for at any depths being influenced by warming from the bed?

Salamatin et al. (2003) report that the temperature dependent activation energy of air diffusion does not change their simulations of air hydrate properties for the Vostok ice core down to a depth of 3300m (temperatures below -13°C). For the GRIP ice core, where the ice temperature increases above -13°C close to the bed, their model fits the data by assuming the air diffusion rate activation energy increases near the bottom.

This means we can assume that there is no or negligible influence on air hydrate growth for ice temperatures colder than at least -13°C. Thus, we mention the possible temperature effect only for the depth region from 2550 m to the bottom.

196 - "No air hydrates were found in lowermost sample". Is this related to the proximity of the bed? Basal temperatures? etc.?

This question is related to a question asked by reviewer#1 (L196). We will provide identical answers to the two related questions:

The lowermost sample is from 2773.91 m depth. This means that it is about 25 cm above the bottom of the ice core. To the best of our knowledge there are no air content measurements for this sample. However, it was reported that, from 2760 m, the concentration of micro inclusions gradually decreases with depth (Faria et al., 2010). The bottom of the core reaches the pressure melting point (-2°C; Faria et al., 2010) and ice core drilling was terminated because subglacial water was entering the borehole (Wilhelms et al., 2014).

Therefore, it could either be accreted ice with no, or very little impurity inclusions (at least in the 10cm sample!). Or meteoric ice that underwent processes which resulted in the expulsion of impurities and gases, i.e. the segregation of impurities to the grain boundary network and the subsequent drainage to the bedrock driven by the hydraulic gradient (Rempel 2002,2005). These processes are considered to alter the basal ice at EDC, Dome Fuji and NEEM (Tison et al., 2015; Ohno et al., 2016; Goossens et al., 2016).

While we currently cannot explain its origin, we believe that the information about the air hydrate content in the deepest sample is an interesting detail that deserves mentioning. Further investigations of the deepest ice at EDML are necessary to determine its origin and explain the absence of air hydrates.

We will add this discussion about the last sample to the end of paragraph 4.3.2

Rempel et al., 2002: https://doi.org/10.1029/2002JB001857

Rempel 2005: https://doi.org/10.3189/172756405781813564

Faria et al., 2010: https://doi.org/10.1016/j.quascirev.2009.10.016

Tison et al., 2015: https://doi.org/10.5194/tc-9-1633-2015

Goossens et al., 2016: https://doi.org/10.5194/tc-10-553-2016

Ohno et al., 2016: https://doi.org/10.1002/2015JF003777

Best regards,

Florian Painer et al.

**EGUSPHERE-2025-633**

**Air clathrate hydrates in the EDML ice core, Antarctica**

**Detailed list of relevant changes**

We changed L15 - L17 from

"Between about 500 m and 1500 m depth, air bubbles gradually transform to clathrate hydrates of air (hereinafter, air hydrates). The beginning and extent of this depth range is commonly referred to as the bubble-hydrate transition zone (BHTZ)."

to:

"In the bubble to hydrate transition zone (BHTZ), air bubbles gradually transform to clathrate hydrates of air (hereinafter, air hydrates)."

We added the reference "Lipenkov V. Y. How air bubbles form in polar ice, Earth's Cryosphere, 2018, 22, 16–28." to L28 – L30.

We added "The maximum logging depth is 2774.15 m (Wilhelms et al., 2014) and the borehole inclination is roughly between  $\pm$  3° (Weikusat et al., 2017)." to L38.

We added information about the extend of the BHTZ at EDML at L41:

"It is dated down to about 2415 m depth, or 145 kyr (Ruth et al., 2007; Bouchet et al., 2023) and the BHTZ extends from 700 m to 1225 m (Bendel et al., 2013)."

Concerning the presentation of the material we followed reviewer #1 advice and moved section 4.1 (L244 – L255) to section 3.1 with slight rewording and including the other comments by reviewer #1 for this paragraph.

3.1.1 Comparison of air hydrate number concentration with air bubble concentration at EDML

Assuming a 1:1 conversion ratio between air bubbles and air hydrates, we estimate the detection rate of our method by comparing our measured Nah with Nab at EDML. Bendel et al. (2013) report an air bubble number concentration (Nab) of 300 - 400 cm-3 for Holocene-ice and 400 - 500 cm-3 for ice formed during the Last Glacial Maximum (LGM). Note that LGM-ice at EDML is located within the BHTZ (at about 1000 m depth), therefore, air bubbles are already converting to air hydrates. Lipenkov (2018) describe a semi-empirical model relating air bubble number and sizes to climate parameters and ice formation conditions (firn temperature, accumulation rate and surface snow density). Using this model we estimate Nab for present-day conditions at EDML (-44.5 °C; 6.4 g cm-2 yr-1; 0.38 g cm-3) to be about 400 cm-3 and Nab in the LGM ice (assuming: -54 °C; 3 g cm-2 yr-1; 0.38 g cm-3) to be about 475 cm-3. We then compare our measured Nah for MIS 5e, as an analogue for Holocene conditions, and MIS 6, to represent the LGM, with Nab. As a result, we detect between 48% and 63% of the expected Nah. One explanation could be that the hydrate-mapping depth in our

microscopic set-up might not correspond to the physical sample thickness (Fig. 1). In other words, microphotographs from one focus plane might not display the entire sample volume.

We also moved section 4.2 (L257 – L262) to 3.2. including the other suggestions by reviewer #1 as follows:

**3.2.1 Comparison of TACh with total air content at EDML**

The total air content (TAC) in polar ice is considered to depend on air pressure, temperature and the pore volume in the firn at the time of pore close-off (e.g. Martinerie et al., 1992). As air hydrates contain most of the ancient air molecules in polar ice (Uchida et al., 2011), we can compare the calculated TACh with other independent TAC estimates. For EDML, a mean TAC of 0.0815 cm3 g -1 was measured by a method integrated to a Continuous Flow Analysis system (Ruth et al., 2004). This average value is lower than the TAC measured with absolute methods in Holocene ice cores from higher elevated drilling sites in Antarctica (Vostok, EDC; Martinerie et al., 1992). Therefore, we compare the calculated TACh to the theoretical TAC expected at EDML for an elevation of 2892 m a.s.l. and an atmospheric pressure of 650 - 700 mbar (about 0.092 cm3 g -1) following Martinerie et al. (1992). The average TACh from 1700 m down to the bottom is 0.059 cm3 g -1. This is about 64% of the expected TAC and matches well with our estimations comparing Nab and Nah. Employing a scaling factor for the Nah of 1.55, the average TACh for this depth region matches the expected TAC value (orange markers in Fig. 7b). Note that this scaling was not applied for the graphs and the data interpretation.

The high TACh values for the depth range from 1255 - 1700 m will be discussed in chapter 4.1.1. The high TACh values for the depth range from and 2382 - 2545 m (region 2) will be discussed in chapters 4.1.2 and 4.2.2

We added L263 – L274 from section 4.2 to section 4.3.1 (now 4.1.1 in the revised version) after L301 and revised the text according to the comments by reviewer #1:

From 1255 m to 1700 m (spanning about 26 kyr), the TACh decreases from 0.085 cm3 g -1 to 0.059 cm3 g -1 (Fig. 7b). For the same depth region, we observe a decrease in the RSD of the air hydrate D distribution (Fig. 5c) and a strong decrease in the AR 90% percentile from 2.5 to 2 (Fig. 8c). One explanation for the overestimation of the TACh (relative to the average value of 0.059 cm3 g -1) could be the fact that the air hydrates' true cage occupancy for this region is probably slightly lower than our estimated 0.9, as it depends on pressure and temperature (Chazallon and Kuhs, 2002). However, the variations in gas hydrate cage occupancies with different formation conditions (e.g. Uchida et al., 1999; Hachikubo et al., 2022) are not large enough to solely explain the observed deviation from the mean. Another reason could be an uncertainty of the air hydrate size, as the calculation of Vc is very sensitive to errors in size determination. In the depth region just below the BHTZ we observe more complex-shaped air hydrates (Fig. 9) compared to deeper parts, which is indicated in the data by the decrease in the RSD and the AR 90% percentile (Fig. 5, 8c). The complexshaped air hydrates partly seem to show a recrystallization towards an equilibrium state (regular and rounded morphologies). For the Vostok ice core, Lipenkov (2000) reported a depth region of faster air hydrate crystal growth rates, extending about 300 m below the BHTZ (i.e. down to 1550 m), and explains this by the large number of small, oxygen-enriched air hydrates dissolving in this region. Down to the same depth, Suwa and Bender (2008) measured noisy δO2/N2 ratios. Similarly, Oyabu

et al. (2021) reported a large scatter in measured  $\delta$ O2/N2 ratios for about 300 m (spanning 25 kyr) below the BHTZ for the Dome Fuji ice core. Therefore, we surmise that the region from about 1255 - 1700 m at EDML coincides with a region of relatively faster air hydrate growth and recrystallization and the air hydrate ensemble might not have reached an equilibrium state. In summary, the overestimation likely originates from uncertainties of measurements in air hydrate size, which increase with complex-shaped air hydrates in the images.

Furthermore, we added a new figure (now Figure 9) displaying examples of complex shaped air hydrates.

We added error-bars to Figure 5b in accordance with a comment by reviewer #1 and revised the figure caption according to suggestions by reviewer #1 and reviewer #2:

Figure 5. Air hydrate depth profiles of the EDML ice core. a) Paleo-climatic information as  $\delta$  180ice values. Blue corresponds to the dated- (EPICA Community Members, 2010), gray to the non-dated part of the ice core (Meyer et al.). b) air hydrate counts and number concentration (Nah). Error-band for air hydrate counts is explained in section 2.4 and Table 2, error-bars are calculated including the uncertainty in sample thickness measurements. c) mean diameter (D) and relative standard deviation (RSD) and d) ice grain size (Weikusat et al., 2009). The ice age at the top of the figure corresponds to the AICC 2023 for EDML (Bouchet et al., 2023). The MIS boundaries are defined by Lisiecki and Raymo (2005), the MIS 5 substage boundaries are from Otvos (2015). Shaded bands correspond to "region 1" and "region 2" discussed in the text (especially in sections 4.2.1 and 4.2.2).

We changed the order of Figure 8 (now Figure 6) as suggested by reviewer #1.

We revised Figure 6 (now Figure 7) based on suggestions by reviewer #1 and reviewer #2 (changed colors, reidentified star markers) and changed the figure caption accordingly:

Figure 7. a) Air hydrate mean volume. The solid line represents the calculated growth rate for the dated part of the EDML ice core (1255 - 2415 m; Ruth et al., 2007). b) Blue markers: air hydrate volume concentration and total air content calculated from theoretical air content of air hydrates. Orange markers: scaled values to match expected mean TAC for EDML (dotted line at 0.092cm3 g -1; Martinerie et al., 1992). Star markers correspond to samples from 1255 - 1294 m depth. Square markers correspond to samples from 2025 - 2115 m depth (region 1 in Figs. 5,8). Triangular markers correspond to samples from 2382 - 2545 m depth (region 2 in Figs. 5,8). The ice age at the top of the figure corresponds to the AICC 2023 for EDML (Bouchet et al., 2023).

We revised the caption of Figure 7 (now Figure 8) according to suggestions by the reviewers:

Figure 8. Shape characteristics of air hydrates in the EDML ice core. a) Paleo-climatic information as  $\delta$  18Oice values. Blue corresponds to the dated- (EPICA Community Members, 2010), gray to the non-dated part of the ice core (Meyer et al.), b) air hydrate median AR and median of the absolute orientations ( $|\alpha|$ ), c) 90% percentile of air hydrate AR and d) measured borehole temperature (Weikusat et al., 2017) and the second eigenvalue of the orientation tensor of the ice crystals c-axis distribution ( $\lambda$ 2) measured on vertical sections (Weikusat et al., 2013). Schematic stereographic

projections are added to display the respective changes in ice CPO indicated by the three "terraces" in  $\lambda 2$ . The complete CPO of EDML is described in Weikusat et al. (2017). The ice age at the top of the figure corresponds to the AICC 2023 for EDML (Bouchet et al., 2023). The MIS boundaries are defined by Lisiecki and Raymo (2005), the MIS 5 substage boundaries are from Otvos (2015). Shaded bands correspond to "region 1" and "region 2" discussed in the text (especially in sections 4.2.1 and 4.2.2).

We added additional information to L289 as commented by reviewer #1

"On the other hand, the correlation between past accumulation and temperature (i.e. stable water isotopes of ice) on smaller time scales is not as obvious as in the case of global glacial-interglacial climate changes. Since the air hydrate number concentration and mean size depend on the temperature and accumulation rate prevailing during the snow to ice transformation (e.g. Spencer et al., 2006; Lipenkov, 2018), this may also be the reason for the weaker correlation between Nah and  $\delta$  180."

We changed L291 – L294 to provide more clarity as commented by reviewer #2

"For samples from 2025 - 2115 m depth (region 1 in Fig. 5), encompassing MIS 5b and parts of MIS 5c, we observe an evident deviation of the climatic influence on the air hydrate ensemble. We measure a low Nah and a relatively large mean diameter (c.f. section 4.4.1), whereas the colder climate during MIS 5b (Fig. 5a) should result in a relatively high Nah and a small average size of air hydrates."

We changed "As a result of the surface to volume ratio, smaller particles are more soluble than larger particles,..."

to

"Due to the greater surface curvature, smaller particles are more soluble than larger particles,..."

In L299-L300.

We revised L312 – L313 according to reviewers#1 comments:

"Furthermore, the emerging of the latter explains the high mean volume and the abnormal volume concentration and TACh (Fig. 7a,b), as the volume of the cluster is overestimated during calculation of the volume from an area assuming a spherical shape. Besides the overestimation in volume, the increase in air hydrate clusters also leads to an underestimation of Nah. However, quantification is challenging due to the difficulty in distinguishing individual air hydrate crystals within clusters from the images."

We revised L326-L327 as suggested by reviewer #1:

"The fundamental reason behind the disappearance of climate-related variations in Nah and D is the fact that air hydrate growth rates are inversely proportional to the cube of their mean initial size (Salamatin et al., 2003). In other words, the growth rates of air hydrate ensembles with smaller mean size are higher than growth rates for ensembles with a larger mean size. Other parameters that influence air hydrate growth are increasing depth (and pressure), increasing age and increasing ice temperature (Uchida et al., 2011)."

We added a paragraph discussing the lack of air hydrates in the lowermost samples as asked by the reviewers. The following was added to L332 in section 4.3.2 (now 4.1.2):

"The reason for the absence of air hydrates in the lowermost sample at a depth of 2773.91 m, that is, about 25 cm above the bottom of the ice core, is currently unclear. The bottom of the core reaches the pressure melting point (-2°C; Faria et al., 2010a) and ice-core drilling was terminated because subglacial water entered the borehole (Wilhelms et al., 2014). Furthermore, Faria et al. (2010a) reported that, from 2760 m, the concentration of micro inclusions gradually decreases with depth. Therefore, the sample could represent accreted ice with no or very little impurity inclusions (at least in the 10 cm sample!). On the other hand, the ice could be of meteoric origin and subject to processes that resulted in the expulsion of impurities and gases, i.e. the segregation of impurities to the grain boundary network and subsequent drainage to the bedrock driven by the hydraulic gradient (Rempel et al., 2002; Rempel, 2005). These processes are considered to alter the basal ice at EDC (Tison et al., 2015), Dome Fuji (Ohno et al., 2016) and NEEM (Goossens et al., 2016). Note that a decrease in TAC was only reported (or measured) for samples of the NEEM ice core (Goossens et al., 2016). Further investigations of the deepest ice at EDML are necessary to determine its origin and explain the absence of air hydrates."

If the reviewers and the editor deem it appropriate, we propose to add a new Appendix C discussing a question raised by reviewer#1 and in parts also by reviewer#2 (e.g. azimuth of ice samples).

This appendix would include 4 new Figures and the following discussion:

**Appendix C: Projection properties of Ellipsoids**

"Figures C1, C2, C3 show the changes in orthogonal projection properties of an ellipsoid that is rotated around the core axis (Z). A and C of the ellipsoid correspond to the major- and minor axis of the projected ellipse and are the same for the three initial conditions (Fig. C1a, C2a, C3a). The third dimension, B, is changed accordingly to represent three end-member cases. One challenge in the interpretation of shape and orientation data lies in the fact that we do not know if we measured the maximum elongation and that the third dimension is unknown, i.e. if the non-isometric object is more cigar (prolate)-like (e.g. Fig. C2) or more oblate-like (e.g. Fig. C3). For the interpretation of the air hydrate shape characteristics, we consider the relative orientation of the ice samples to each other. As mentioned in section 2.5.2, a change in relative orientation could be caused due to a difficulty in matching core breakpoints during ice-core logging. Weikusat et al. (2017) report a loss in azimuthal orientation of the core between 1686 m and 1696 m as well as between 1955 m and 2035 m depth. Between 1665 m and 1695 m, we observe an increase in air hydrate median orientation from 17° to 22° (Fig. 8b) as well as a "polarity" change (Fig. C4a,b). However, we do not observe a significant change in the median AR (Fig. 8b). Between 2005 m and 2035 m depth, we observe a

more continuous decrease in the median orientation (Fig. 8b and Fig. C4c,d,e) and an increase in the median AR (Fig. 8b). The fact that the median AR does not change between 1665 m and 1695 m, while the orientation does, implies that either the air hydrates are not very well sorted or that the majority is rather isometric in shape. On the contrary, the continuous and joint change of AR together with orientation from 2005 m to 2035 m depth point either towards an increase in sorting or a decrease in isometric air hydrates. Both could be induced by the increase in simple shear deformation. Future work should consider taking images from two (or more) different directions to increase the accuracy in air hydrate shape and orientation measurements."

**Other technical corrections:**

- We revised the resolution of Figure 9 (now Figure 10).
- We added the following statement to Code and data availability (Related to a comment by reviewer #2): "The  $\delta$  180ice values for EDML from 2415 m to the bottom will be published in PANGAEA by Meyer et al.."
- We extended the Acknowledgements: "This work is a contribution to the European Project for Ice Coring in Antarctica (EPICA), a joint European Science Foundation/European Commission scientific programme, funded by the European Union (EPICA-MIS) and by national contributions from Belgium, Denmark, France, Germany, Italy, the Netherlands, Norway, Sweden, Switzerland and the United Kingdom. The main logistic support was provided AWI at Dronning Maud Land. This work is EPICA publication no. XXX."

Best regards,

Florian Painer et al.